



# A derecho climatology (2004-2021) in the United States
# based on machine learning identification of bow echoes
Jianfeng Li[1, *], Andrew Geiss[1], Zhe Feng[1, *], L. Ruby Leung[1], Yun Qian[1], Wenjun Cui[2, 3]
[1]Atmospheric, Climate, and Earth Sciences Division, Pacific Northwest National Laboratory,
Richland, Washington, USA
[2]Cooperative Institute for Severe and High-Impact Weather Research and Operations, University
of Oklahoma, Norman, Oklahoma, USA
[3]National Severe Storms Laboratory, National Oceanic and Atmospheric Administration,
Norman, Oklahoma, USA
*Correspondence to Jianfeng Li (jianfeng.li@pnnl.gov) and Zhe Feng (zhe.feng@pnnl.gov)



## Abstract

Due to their persistent widespread severe winds, derechos pose significant threats to human safety and property, and they are as hazardous and fatal as many tornadoes and hurricanes. Yet, automated detection of derechos remains challenging due to the absence of spatiotemporally continuous observations and the complex criteria employed to define the phenomenon. This study proposes a physically based definition of derechos that contains the key features of derechos described in the literature and allows their automated objective identification using either observations or model simulations. The automated detection is composed of three algorithms: the Flexible Object Tracker algorithm to track mesoscale convective systems (MCSs), a semantic segmentation convolutional neural network to identify bow echoes, and a comprehensive algorithm to classify MCSs as derechos or non-derecho events. Using the new approach, we develop a novel high-resolution (4 km and hourly) observational dataset of derechos over the United States east of the Rocky Mountains from 2004 to 2021. The dataset is analyzed to document the derecho climatology in the United States. Many more derechos (increased by ~50-400%) are identified in the dataset (~31 events per year) than in previous estimations (~6-21 events per year), but the spatial distribution and seasonal variation patterns resemble earlier studies with a peak occurrence in the Great Plains and Midwest during the warm season. In addition, around 20% of damaging gust ($\geq$ 25.93 m s$^{-1}$) reports are produced by derechos during the dataset period over the United States east of the Rocky Mountains. The dataset is available at https://doi.org/10.5281/zenodo.10884046 (Li et al., 2024).

# 1 Introduction

A derecho is qualitatively defined as a widespread, long-lived straight-line windstorm associated
with a fast-moving mesoscale convective system (MCS). Figure 1 shows two of the most destructive
derechos in the United States: the June 2012 North American derecho and the August 2020 Midwest
Derecho. Both events lasted for over 10 hours, with apparent bow echoes and extensive damaging wind
gusts ($\geq$ 25.93 m s$^{-1}$). Due to the persistent widespread damaging gusts, derechos can severely damage
property and threaten human lives, as exemplified by the extensive power outages and more than ten
fatalities caused by the two derechos. Ashley and Mote (2005) demonstrated that derechos could be as
hazardous as and were comparable in magnitude to most hurricanes and tornadoes in the United States
between 1986 and 2003.

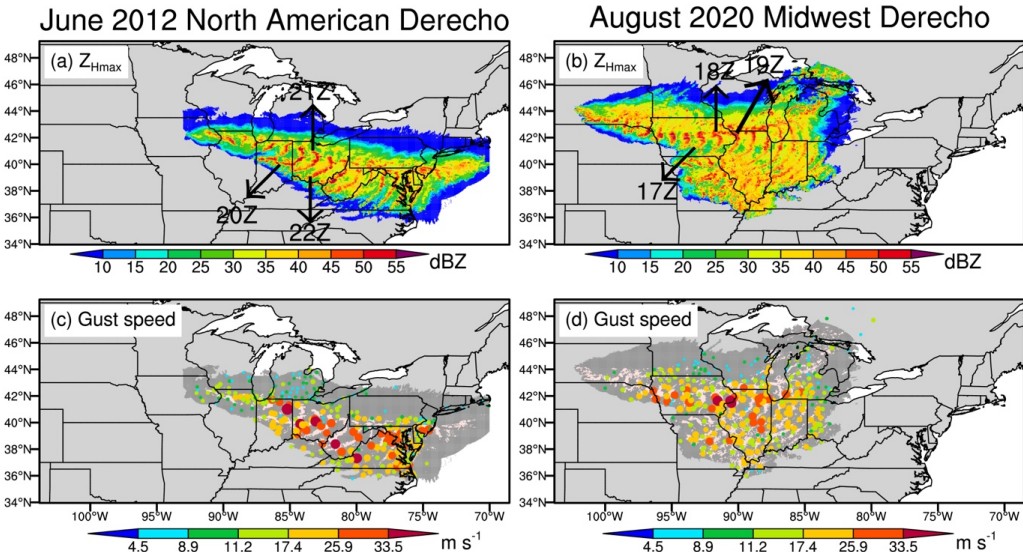

Figure 1. Spatial evolutions of the (a, b) composite (column-maximum) radar reflectivity ($Z_{Hmax}$) signatures
and (c, d) surface gust speeds (colored dots) of two derechos. The first column is for the June 2012 North
American derecho, which occurred on 29-30 June 2012, and the right column is for the August 2020 Midwest
derecho, which occurred on 10-11 August 2020. Due to spatiotemporal overlapping, multiple $Z_{Hmax}$ and gust
speeds may exist for a given grid cell or weather station, in which case only the corresponding maximums are
shown in the figure. The timings of some bow echo occurrences are labeled in (a) and (b). In (a), "20Z",
"21Z", and "22Z" refer to 20:00, 21:00, and 22:00 UTC on 29 June 2012. In (b), "17Z", "18Z", and "19Z"
refer to 17:00, 18:00, and 19:00 UTC on 10 August 2020. In (c) and (d), the misty rose shading corresponds to
areas with $Z_{Hmax} \geq$ 40 dBZ, and the dark gray shading refers to derecho coverage with $Z_{Hmax} <$ 40 dBZ. The dot



sizes in (c) and (d) are proportional to the gust speed magnitudes. Notably, gust speed in (c) and (d) is based on
the hourly maximum gust speed ($gust_{hourly\_max}$), which is the largest gust speed within one hour if multiple gust
speed measurements are available.
A reliable derecho dataset is foundational for understanding the underlying physical mechanism of
derecho initiation and development and their socioeconomic impacts. Johns and Hirt (1987) developed
the first derecho climatology in the warm seasons of 1980-1983 in the United States by quantitatively
defining a derecho as a family of downburst clusters produced by an extratropical MCS. Specifically, they
required a derecho to satisfy the following six criteria. 1) There must be a concentrated area of reports
with wind damage or convective gusts > 25.7 m s$^{-1}$, with a major axis length of at least 400 km. 2) These
reports must show a pattern of chronological progression, either as a singular swath or a series of swaths.
3) The concentrated area must have at least three reports of either F1 damage (32.7-50.3 m s$^{-1}$) (Fujita,
1971) or convective gust of at least 33.4 m s$^{-1}$ separated by $\geq$ 64 km. 4) At most 3 hours can elapse
between successive reports of wind damage or gust > 25.7 m s$^{-1}$. 5) The associated convective system
must have temporal and spatial continuity in surface pressure and wind fields. 6) If multiple swaths of
wind damage or gust reports > 25.7 m s$^{-1}$ exist, they must be from the same MCS event. Since then,
several other studies have developed derecho climatologies during other periods using slightly different
criteria (Bentley and Mote, 1998; Evans and Doswell, 2001; Bentley and Sparks, 2003; Coniglio and
Stensrud, 2004; Guastini and Bosart, 2016). For example, Bentley and Mote (1998) removed the third
requirement and reduced the elapsed time in the fourth condition from no more than 3 hours to no more
than 2 hours in their derecho climatology from 1986 to 1996. In Coniglio and Stensrud (2004), the
elapsed time was further changed to no more than 2.5 hours, and the gust reports of at least 33 m s$^{-1}$ were
used to separate derechos of different intensities.
Although the aforementioned derecho datasets were generated using different criteria and during
different periods (Johns and Hirt, 1987; Bentley and Mote, 1998; Evans and Doswell, 2001; Bentley and
Sparks, 2003; Coniglio and Stensrud, 2004; Guastini and Bosart, 2016), they showed many similar
derecho climatological characteristics in the United States. For example, derechos occur more frequently





in the warm than cold seasons; the Great Plains, Midwest, and Ohio Valley are regions most favorable for
derecho development, and few derechos occur in the eastern and western coastal areas. Considering the
inconsistent thresholds used in the above studies and the lack of physical mechanisms in their derecho
definitions, Corfidi et al. (2016) proposed a stricter and more physically based derecho definition, which
required the existence of sustained bow echoes with mesoscale vortices or rear-inflow jets and a nearly
continuous wind damage swath of at least 100 km wide along most of its extent and 650 km long. In
addition, the wind damage must occur after the convective system was organized into a cold-pool-driven
forward-propagating MCS. Most derechos satisfying this definition would be classified as "progressive"
but not "serial." A serial derecho typically originates in strongly forced environments and develops from a
mature squall line with multiple embedded bow echoes. In contrast, progressive derechos generally
originate from small convective clusters that grow upscale into large organized forward-propagating
MCSs in synoptic environments with weak forcing (Squitieri et al., 2023).
It is difficult to develop a derecho climatology using the definition proposed by Corfidi et al. (2016)
with current operational measurements, as it involves the identification of bow echoes, rear-inflow jets,
and cold pools. However, rear-inflow jets and cold pools are generally associated with bow echoes
(Weisman, 1993; Adams-Selin and Johnson, 2010). Once long-lived bow echoes are found in an MCS
event, we can expect the simultaneous existence of rear-inflow jets and cold pools. Nevertheless,
identifying bow echoes, a feature typically identified from radar observations, is still challenging for large
volumes of data, such as the 30+ year National Oceanic and Atmospheric Administration (NOAA) Next
Generation Weather Radar (NEXRAD) archive. The manual examination is time-consuming and sensitive
to subjective biases. This study applies a semantic segmentation convolutional neural network (CNN) to
detect bow echoes automatically from two-dimensional composite (column-maximum) reflectivity ($Z_{Hmax}$)
data in the United States, which are then combined with an MCS tracking dataset and gust speed
measurements from surface meteorological stations to identify derechos using criteria adjusted from
Corfidi et al. (2016). After manual examination and validation, we produce a high-resolution (4 km and



hourly) observational derecho dataset in the United States east of the Rocky Mountains from 2004 to
2021. As the first derecho climatology that utilizes a machine learning technique following physically
based criteria and covers the recent decades, the dataset provides a reference for future derecho studies
and can be used to investigate the underlying mechanisms for derecho initiation and development, the
climatological impacts of derechos on hazardous weather, and the damage of derechos to infrastructure
and human property.
The remainder of the paper is organized as follows. Section 2 introduces the MCS dataset and gust
speed measurements used to generate the derecho dataset. Section 3 describes the machine learning (i.e.,
semantic segmentation CNN) methodology to detect bow echoes, including sampling, training, and
evaluation. Section 4 explains our derecho identification criteria in detail. Section 5 evaluates our derecho
dataset against the observational data from the NOAA Storm Prediction Center (SPC) in 2004 and 2005.
Section 6 analyzes the derecho climatological characteristics. Section 7 shows how to access our derecho
dataset, and the study is summarized in Section 8.

## 116 2 Source datasets

### 117 2.1 MCS dataset

Since previous MCS datasets only cover the period from 2004 to 2017 (Li et al., 2021; Feng et al.,
2019), we use an updated version of the Python FLEXible Object TRacKeR (PyFLEXTRKR) software
(Feng et al., 2023), which exploits collocated radar signatures, brightness temperature, and precipitation
to identify robust MCS events (Feng et al., 2019), to produce an updated MCS dataset in the United States
east of the Rocky Mountains from 2004 to 2021. Several source datasets are used in the generation of the
MCS dataset, including the National Centers for Environmental Prediction (NCEP)/the Climate
Prediction Center (CPP) L3 4 km Global Merged IR V1 brightness temperature dataset (Janowiak et al.,
2017), the three-dimensional Gridded NEXRAD Radar (GridRad) dataset (Bowman and Homeyer, 2017),



the NCEP Stage IV precipitation dataset (CDIACS/EOL/NCAR/UCAR and CPC/NCEP/NWS/NOAA,
2000), and hourly melting level heights derived from ERA5 (European Centre for Medium-Range
Weather Forecasts (ECMWF) Reanalysis v5) (Hersbach et al., 2023). The MCS definition criteria are
almost the same as those in Feng et al. (2019), such as cold cloud shield (CCS) area > 60,000 $km^2$,
precipitation feature (PF, which is a continuous convective or stratiform area with surface rain rate > 2
mm $h^{-1}$) major axis length > 100 km, the existence of 45-dBZ convective echoes, etc., except that the
duration requirement is lowered to include those convective systems lasting for just 6 hours. This
adjustment allows us to capture slightly shorter-lived MCSs that produce intense wind gusts but are
missed in the previous MCS datasets. Convective and stratiform radar echo classification in
PyFLEXTRKR follows the Storm Labeling in 3D (SL3D) algorithm (Starzec et al., 2017), which uses
horizontal texture and vertical structure of radar reflectivity from the GridRad product. Notably, the
GridRad data are available each month from 2004 to 2017 but only between April and August from 2018
to 2021. Since most derechos occur in the warm season (Ashley and Mote, 2005; Coniglio and Stensrud,
2004), missing the cold months between 2018 and 2021 does not affect our derecho climatological
analyses in Section 6. For brevity, we do not mention the missing cold months between 2018 and 2021 in
the following sections unless stated otherwise.
**2.2 Surface gust speed observations**
Surface gust speed measurements between 2004 and 2021 are from the Integrated Surface Database
(ISD) (NOAA/NCEI, 2001), developed by the NOAA National Centers for Environmental Information
(NCEI) in collaboration with several other institutions. ISD compiles global hourly and synoptic surface
observations from numerous sources (e.g., the Automated Surface Observing System and the Automated
Weather Observing System) into a single common format and data model. Besides internal quality control
procedures conducted by the source datasets, ISD applies additional quality control algorithms to process
each observation through a series of validity checks, extreme value checks, and internal and external





continuity checks (Smith et al., 2011). This study uses measurements passing all quality control checks
(NOAA/NCEI, 2018). Notably, there may be multiple measurements at different times within one hour
for some stations. To keep the sampling consistency across different datasets used in the derecho
identification, we calculate $gust_{hourly\_max}$, which is the largest gust speed of all available measurements
within one hour, for each observational site, unless stated otherwise. A total of 4,260 observational sites
provide gust speed measurements between 2004 and 2021 in the study domain, of which 3,954 are over
land, and the rest are over the ocean or lakes (Figure S1). We have excluded one observational site (ISD
station ID: 726130-14755) in the northeastern United States, which has an unrealistic number of
damaging gust measurements (more than 1,000 hours), inconsistent with the surrounding sites. We note
that although we only use measurements passing all the available quality control checks, spatial quality
control is missing in the ISD (Smith et al., 2011). Figure S2a shows that some sites in the eastern United
States have apparently more damaging gust occurrences than their surrounding sites, but the occurrence
frequencies are less than those stations around the Rocky Mountains. We do not have enough evidence to
exclude them from the study. However, the quality of the gust speed measurements will undoubtedly be a
source of uncertainty for our derecho dataset. In addition, only 420 sites have continuous gust
measurements from 2004 to 2021, while the rest have gust measurements only during part of the study
period. The availability of observational sites is another source of uncertainty when identifying derechos.

## 3 Machine learning identification of bow echoes

A bow echo is a bow-shaped pattern on a radar image, but its vague definition makes it hard to

identify them extensively and efficiently using traditional methods. Instead, we train a semantic
segmentation CNN to identify bow echoes automatically from two-dimensional $Z_{Hmax}$ images by
performing pixel-level labeling of the bow echo extent. Compared to the manual examination of radar
images, machine learning can save a tremendous amount of time and eliminate subjective bias.

## 3.1 Bow Echo Samples

### 3.1.1 Initial manual sampling

Our initial bow echo samples are generated based on the named derechos on Wikipedia
(https://en.wikipedia.org/wiki/List_of_derecho_events; last access: 19 March 2023). We identify 54
named derechos in the MCS dataset and manually label times with apparent bow echoes through visual
inspection of $Z_{Hmax}$ associated with the tracked MCSs. Each positive sample is a 384 × 384-pixel (~1536
km × 1536 km) $Z_{Hmax}$ image centered at the corresponding derecho with a bow echo embedded (Figure 2).
The number of bow echo samples varies among different derechos, and 566 positive samples are obtained
in total. 5400 negative samples are also randomly selected from the radar reflectivity dataset.

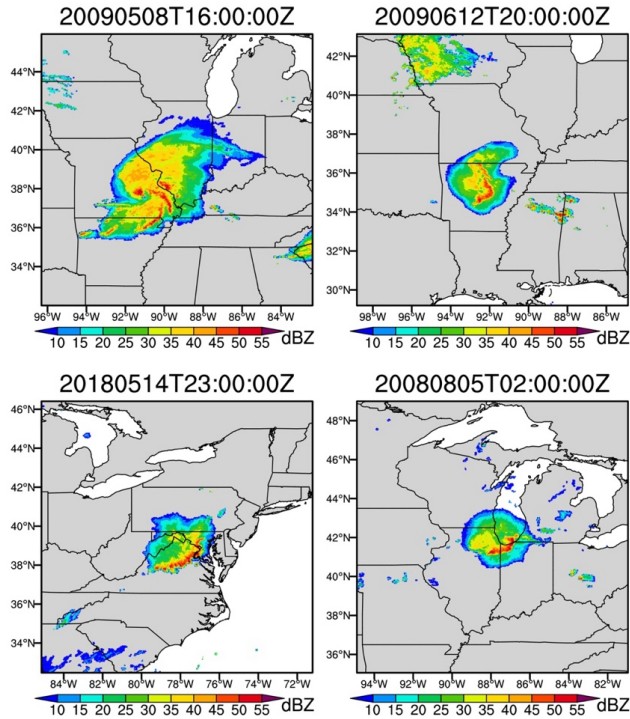

Figure 2. Four examples of bow echoes from the named derechos. The color shading is for $Z_{Hmax}$. The subplot
titles indicate the bow echo timings. For example, 20130613T04:00:00Z represents 4:00 UTC on 13 June
2013.





*3.1.2 CNN-based selection of additional bow echo samples*

Our initial attempt at developing an automated bow echo detection scheme is to train a classifier

CNN — "Dense Net" (Huang et al., 2019) that ingests 384×384-pixel single-channel $Z_{Hmax}$ images and
outputs a single classification indicating the presence of a bow echo. Dense Nets are notable for their
large number of skip connections, and they can achieve comparable performance to very large classifier
CNNs with only a fraction of the trainable parameters. Unfortunately, a Dense Net trained on the
aforementioned initial samples has a very high false positive rate when applied to the full radar dataset
(determined by manual inspection). Although this Dense Net is unsuitable for deployment, the collection
of new positive samples it successfully identifies allows us to supplement the list of known bow echoes
and develop a more diverse training set for the following segmentation model.
*3.1.3 Pseudo-labeling*

By combining the initial samples and the manually selected true positives from the low-quality

Dense Net model, we build a semantic segmentation training dataset of 500 unique bow echo snapshots
and corresponding hand-drawn bow echo masks. While 500 positive samples are relatively small for a
deep learning application, these samples have higher diversity than the initial bow echoes generated from
the named derechos on Wikipedia because they are drawn from more distinct events, and, in general,
semantic segmentation CNNs can be successfully trained with far fewer samples than image classification
CNNs (Bardis et al., 2020).

A relatively low-skill version of the semantic segmentation CNN is trained using the 500 hand-

labeled radar images and then applied to the entire $Z_{Hmax}$ dataset. We manually review the bow echo
masks produced by this segmentation model and add some of the high-quality masks to a new training
dataset. We also collect some of its false positive masks as new negative samples in the new training
dataset. This is a semi-supervised learning approach known as "pseudo-labeling" or "bootstrapping" (Van



Engelen and Hoos, 2020; Ouali et al., 2020) and is commonly applied to semantic segmentation problems
because of the high expense of hand-drawn labels (Peláez-Vegas et al., 2023). The pseudo-labels and
hand-labels are combined into a final training dataset with 3677 samples, including 1699 bow echoes and
1978 negative samples, which is used to train the much more skillful semantic segmentation model in
Section 3.2.
*3.1.4 Data augmentation*
To combat the limited training data further, we use several data augmentation strategies when
constructing training batches. During training, positive and negative samples are selected with equal
probability, and a batch size of 8 is used. First, random salt and pepper noise is added to 10% of the pixels
in each sample with a probability of 0.1. Second, weak random Gaussian noise with a standard deviation
of 5 dBZ is added to all the pixels in each sample with a probability of 0.1. Third, samples are flipped in
up-down and left-right directions, each with a likelihood of 0.5. Fourth, samples are rotated by 0, 90, 180,
or 270 degrees, each with a probability of 0.25. Fifth, samples are randomly shifted vertically and
horizontally by -5 to 5 pixels. Sixth, the brightness of the sample image is adjusted by a random factor of
-0.6 to +0.2, and the image contrast is randomly adjusted by -0.2 to 0.2. Seventh, the binary target bow
echo masks are multiplied by 0.9, and random noise drawn from a uniform distribution between 0 and 0.1
is added. This is known as "soft labels." Lastly, both positive and negative samples are blended with
randomly selected negative samples by taking the pixel-wise maximum reflectivity values of the two
samples with a 0.5 likelihood. This last data augmentation is unusual but works well in our application
because a) reflectivity features typically occupy only a fraction of the sample area, with most pixels being
clear-sky and b) bow echoes are high-reflectivity features. When the last data augmentation is applied to a
positive sample, the resulting image will typically still contain a bow echo that matches the target mask
well.



## 3.2 Training of U–Net 3+ CNN


Our final semantic segmentation CNN model (Figure 3) uses the U-Net 3+ architecture (Huang et
al., 2020). U-Net 3+ is a modern variant of the U-Net architecture (Ronneberger et al., 2015) and differs
from the U-Net primarily in the addition of many more skip connections and its multi-resolution loss,
which computes loss on rescaled classification masks generated from feature representations at various
model levels.
U-Net models were originally developed for the segmentation of biomedical imagery but have been
applied to image segmentation problems in other fields and are broadly useful for any image-to-image
mapping tasks where the input and target data are the same (or similar) size and shape and merging multi-
resolution information from the input data is important. U-Net CNNs have been applied to a myriad of
problems in the atmospheric sciences, such as segmentation (Galea et al., 2024; Kumler-Bonfanti et al.,
2020), super resolution (Geiss and Hardin, 2020; White et al., 2024), physics parameterization
(Lagerquist et al., 2021), downscaling (Sha et al., 2020), and weather forecasting (Weyn et al., 2021).
Perhaps most closely related to this study is Mounier et al. (2022), who used a U-Net to detect bow
echoes in simulated radar reflectivity images from a forecast model. A U-Net is an appropriate choice for
the segmentation of bow echoes because merging multi-resolution information is crucial for identifying
the feature. For example, bow echoes have high reflectivity at the pixel scale, strong reflectivity gradients
in the transverse direction at the mid-scale (tens of pixels), and the characteristic bow shape at the large
scale (hundreds of pixels).

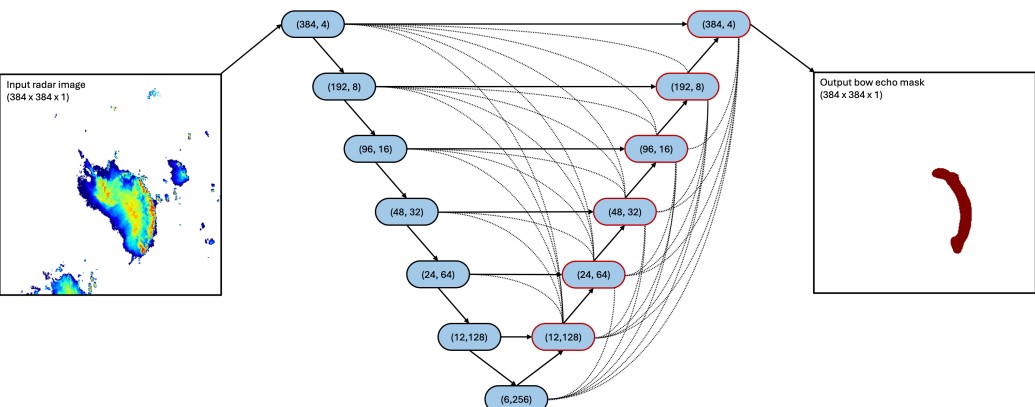

Figure 3. A diagram of our semantic segmentation CNN architecture. The CNN inputs a 384×384-pixel radar
image ($Z_{Hmax}$ scaled to 0-255) and outputs a bow echo mask of the same size. The blue ovals represent 3×3
convolutional layers, each followed by a batch normalization layer and a leaky rectified linear unit (ReLU)
activation function. The first number in each blue oval indicates the spatial size (for both the width and height)
of the output tensor, and the second represents the number of output channels. The solid arrows indicate
connections in a standard U-Net architecture, with the downward arrows corresponding to 2×2 max-pooling
and the upward arrows corresponding to 2×2 bilinear upsampling operations. The dashed lines represent the
skip connections introduced in the U-Net 3+ architecture. These skip connections use max-pooling for spatial
downsampling and bilinear interpolation for upsampling, followed by a 16-channel 3×3 convolutional layer
with a linear activation. Layers with multiple inputs use channel-wise concatenation to combine those inputs.
During training, the output tensors from the layers in the upsampling branch (blue ovals with red boundaries)
are scaled to the output spatial resolution and passed through a 1-channel 1×1 convolutional layer with sigmoid
activation. Training loss is computed on all 6 of the resulting masks. At inference time, only the mask
outputted from the upper-rightmost layer is used.

Our U-Net 3+ CNN ingests 384×384-pixel $Z_{Hmax}$ images where $Z_{Hmax}$ have been clipped to a 0-

50dBZ range and then linearly mapped to a range of 0-255. It is trained using binary cross entropy loss on
masks generated from its 384, 192, 96, 48, 24, and 12-pixel resolution feature representations (Huang et
al., 2020), though only the full-resolution (384×384-pixel) output mask is used at inference time. A
detailed diagram of the model architecture is shown in Figure 3. Notably, although the model is trained
using 384×384-pixel samples, it is a fully convolutional model and can process inputs of variable sizes.

We use the Adam optimizer (Kingma and Ba, 2014) with the Keras default settings (Ketkar, 2017)

and an initial learning rate of 0.001 for training. The U-Net 3+ CNN is first trained for 60 epochs
composed of 1000 randomly generated training batches of 8 samples each. Then, we decrease the learning





rate to 0.0001 and train the CNN for an additional 20 epochs. The training duration is determined by
performing an initial 5 rounds of training with 5-fold cross-validation and approximating the epoch
numbers to reduce the learning rate and stop training when the mean intersection over union metric
plateaus for the validation set. Instead of random shuffling, the validation sets are separated from the
training dataset in temporally contiguous chunks to avoid any overlap because, sometimes, multiple
samples may be drawn from different times of the same convective system.

## 3.3 Evaluation of the Semantic Segmentation CNN

We apply the trained U-Net 3+ CNN to the entire $Z_{Hmax}$ dataset and obtain potential bow echo masks
over the United States between 2004 and 2021 (Figure 4). As a final post-processing step, we ignore
"bow echo" masks with less than 20 pixels (~320 km$^2$), which are too small to be classified as bow
echoes.

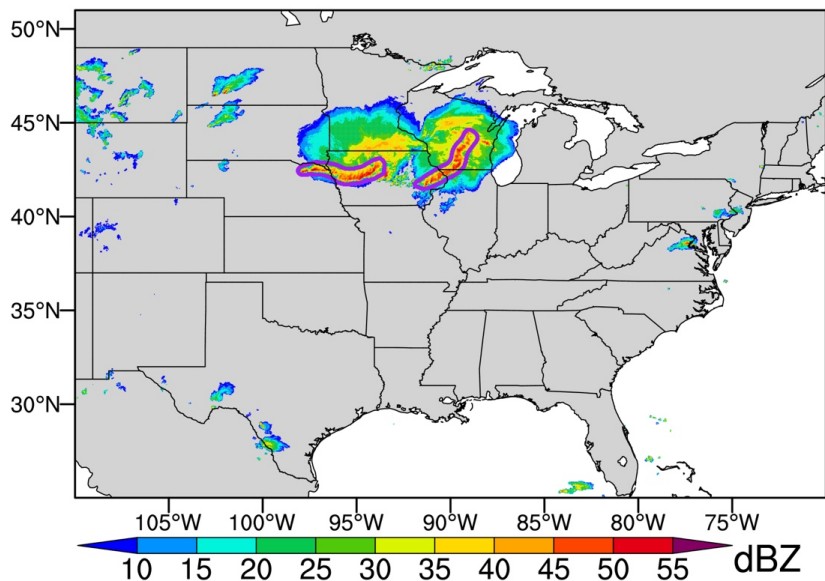

Figure 4. Examples of the U-Net 3+ CNN identified bow echoes (purple contours) based on $Z_{Hmax}$ (color
shading) at 5:00 UTC on 17 June 2014.



Instead of validating our segmentation model at a pixel scale, as during the training stage, we prefer
evaluating its performance in detecting bulk bow echo features. In other words, we care about whether the
segmentation model can recognize the existence of bow echoes and capture their rough locations. Minor
spatial biases in bow echo coverage do not affect our below derecho identification, which contains
various flexible criteria to minimize their impacts, such as the buffer zone within 100 km of bow echoes.
We also choose to validate the segmentation CNN specifically on MCS events where high reflectivity
features are present. Identifying low-reflectivity and clear-sky images as non-bow echoes is desirable for
our segmentation model but trivial and not of particular interest for creating a derecho climatology.
To build a testing dataset, we randomly select 217 MCS-associated $Z_{Hmax}$ images in 2010 based on
the following requirements. Each image is from a different MCS event. The images have variable sizes
and contain the full spatial extents of the MCSs at the selected times; however, they must be at least
192x192 pixels and cannot be drawn from a day that also has a sample in the training dataset. Three of the
authors independently assessed the presence of bow echoes in each image, the results of which are then
compared to the segmentation CNN (Table 1). Overall, the CNN model identifies 57 bow echoes, while
human labelers 1, 2, and 3 identify 46, 76, and 66, respectively. The average human-human agreement
and $F_1$ scores are 82% and 0.69, while the average human-CNN agreement and $F_1$ scores are 82% and
0.67 (Table 1). The test indicates that, on the one hand, the detection of bow echoes in radar images is
prone to subjective bias; on the other hand, the performance of the segmentation CNN is comparable to
that of a human in identifying bow echoes. We emphasize that the CNN bow echo identification is only
one component in our following derecho detection criteria, and the adverse impact of this uncertainty is
mitigated by other constraints (e.g., almost continuous bow echo existence and strong gusts in proximity
with bow echoes).
Table 1 Evaluation of the performance of the segmentation CNN in the bow echo identification[1]

|  | CNN (57[2]) | Person 1 (46) | Person 2 (76) | Person 3 (66) |
|---|---|---|---|---|
| CNN |  | 84% | 79% | 83% |
| Person 1 | 0.66 |  | 77% | 88% |





| | | | |
|---|---|---|---|
| Person 2 | 0.66 | 0.59 | 81% |
| Person 3 | 0.70 | 0.77 | 0.70 |

[1]The upper part of the table shows agreement between two independent identifications ($Agreement = \frac{TP+TN}{TP+TN+FP+FN}$), and the lower part shows $F_1$ scores ($F_1 = \frac{2TP}{2TP+FP+FN}$), which is a better indication of the ability to agree on positives when positives are a minority (Taha and Hanbury, 2015). Here, $TP$ denotes true positive, $TN$ refers to true negative, $FP$ is false positive, and $FN$ is false negative. Notably, for the comparison between any two independent identifications, we consider one as "true" and evaluate the other against it (and which set of classifications are considered true does not impact these two metrics).
[2]The number of identified bow echoes from the 217 images.

We match the segmentation CNN detected bow echoes with MCS events from the MCS dataset and identify those MCS-associated bow echoes, which are used to identify derechos in the following section. Figure 5 shows the spatial distribution of MCS-associated bow echo occurrences from 2004 to 2021, which is similar to the MCS spatial distribution with more occurrences in the Great Plains (Li et al., 2021).

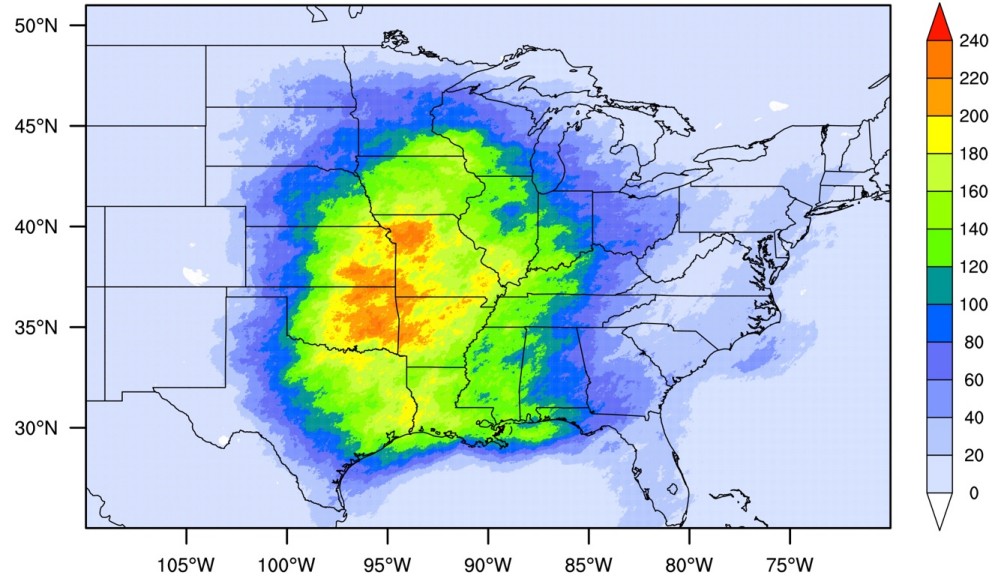

Figure 5. Spatial distribution of the number of MCS-associated bow echoes from 2004 to 2021. Here, we use bow echo masks produced by the segmentation CNN and exclude bow echoes that do not overlap with MCS events. Notably, the PyFELXTRKR-generated MCS dataset contains tropical cyclones (TCs). This figure excludes bow echoes from those non-derecho MCS events that overlap with TCs from the International Best Track Archive for Climate Stewardship (IBTrACS) Version 4 data over the North Atlantic basin (Knapp et al., 2010) following the approach of (Feng et al., 2021).



## 4 Derecho identification


As mentioned above, we adopt the derecho definition proposed by Corfidi et al. (2016) but revise
some criteria based on previous studies (Johns and Hirt, 1987; Bentley and Mote, 1998) and the limitation
of the observational datasets used in this study so that they can be used in the objective identification of
derechos. Our detailed definition criteria are summarized below.
1) A derecho must be attached to an MCS from the MCS dataset. This is the most straightforward
requirement and one of our advantages. Due to the lack of a reliable MCS dataset, most previous
studies spent much effort identifying spatiotemporally continuously propagating convective
systems (Squitieri et al., 2023).
2) At least one derecho feature (DF) exists in the MCS lifetime. A DF is defined as a continuous
period satisfying the following criteria (Figure 6).
2.1) The DF must last for at least 2 hours, and bow echo occurs during ≥ 80% of the DF period.
For example, if the DF lasts for 10 hours, it must have bow echoes in at least 8 hours. In
addition, no more than 2 hours can elapse between successive bow echo occurrences. In other
words, bow echo must exist for at least one hour in any two consecutive hours. The above two
thresholds consider the segmentation CNN identification uncertainty and the diversity of MCS
events. Moreover, a DF requires these bow echoes to be from the same bow echo series. Due to
merging or splitting or the complex nature of some convective systems, a bow echo at one hour
may be far from the bow echoes right after or before that hour or another bow echo during that
hour (Figure 6). In such a rare situation, these bow echoes are unlikely caused by the same
physical process and, therefore, do not belong to the same bow echo series. We separate
different bow echo series in two steps. First, a distance criterion categorizes multiple bow echoes
in the same hour into different series. Any bow echoes more than 100 km from each other
belong to different series. Second, we temporally connect bow echoes from the same series using

Earth System Discussions
Open Access Science Data

another distance threshold. The distance between two successive bow echoes (no more than 2
hours can elapse between their occurrences) from the same series must be no more than 200 km.
Notably, the second step can overwrite the first step. For example, two bow echoes at hour $t$
belong to different series in the first step, but in the second step, they are close enough ($\leq 200$
km) to the same bow echo at hour $t$-1. If so, they are considered from the same bow echo series.

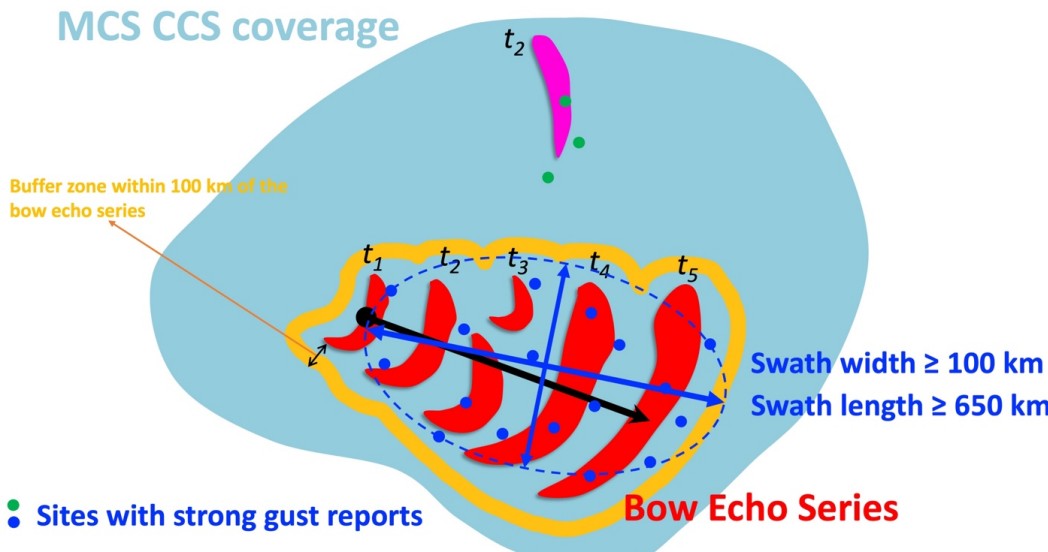

Figure 6. Schematic of the automated detection algorithm. Red and pink contours represent bow echoes. At
time $t_2$, there are two bow echoes belonging to different bow echo series due to their great distance from each
other. In contrast, the two bow echoes at $t_3$ are from the same bow echo series since they are close to each
other. The pink bow echo at $t_2$ is far from the bow echoes at $t_1$ and $t_3$. Therefore, they belong to different bow
echo series. The sites (green dots) with strong gust reports outside the 100-km buffer zone of the bow echo
series (i.e., the DF area) are excluded from the strong gust swath calculation. The black arrow indicates the
propagation direction of the bow echo series.
2.2) We calculate the DF-associated maximum gust speed for each land observational site during
the DF period. Within 100 km of the DF bow echoes, which we name the DF area, there must be
$\geq 10$ sites with strong gusts (gust speed $\geq 17.43$ m s$^{-1}$) and $\geq 1$ site with damaging gusts (gust
speed $\geq 25.93$ m s$^{-1}$). In addition, the fraction of sites with strong gusts should be $\geq 20\%$. This
fraction criterion is intended to exclude potential MCSs associated with extratropical cyclones,
which could produce strong or damaging gusts over limited observational sites but weaker gusts

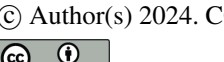



at most other sites. Besides, a DF requires that no more than 2 hours can elapse between
successive strong gust reports. Then, we calculate the major and minor axis lengths of the fitted
ellipse swath using the locations of those sites with strong gust reports (Figure 6). As a DF, the
major and minor axis lengths must be at least 650 km and 100 km, respectively. We emphasize
that our gust speed criteria are weaker than those of previous studies (Squitieri et al., 2023;
Bentley and Mote, 1998; Johns and Hirt, 1987), which estimated the gust swath based on
damaging gusts. Moreover, previous studies often required a few reports of gust speed $\geq 33$ m s$^{-1}$
. Notably, many gust speeds in earlier studies were from post-disaster estimates, while this
study uses ISD surface station measurements. Post-disaster estimates can capture damaging gust
occurrences over a much larger area. In contrast, due to the limited coverage of observational
sites, real-time measurements may miss substantial damaging gust occurrences in nearby
regions. Therefore, we lower the gust speed criteria to capture potential derechos.
2.3) If no DF is identified for a given MCS using the above procedures, we can relax the
distance requirement in (2.2) to be within 200 km of the DF bow echoes that satisfy the
condition that there is no bow echo from the same bow echo series an hour ago or later during
the DF period. If the bow echo is in the first hour of the DF period and there are no CNN-
identified bow echoes for the MCS event an hour ago, we can also extend the distance threshold
to 200 km. This is similar to the bow echo in the last hour of the DF period but without any
CNN-identified bow echoes an hour later. Notably, the distance extension is optional. For the
bow echoes satisfying the above conditions, the distance threshold can be either 100 or 200 km.
Using 100 km is superior to using 200 km until we find a DF if it exists. The distance extension
is also intended to minimize the impacts of the bow echo identification error. If a bow echo is
missed in the semantic segmentation procedure, extending the distance threshold can include
strong gusts associated with the missed bow echo, thus slightly reducing the derecho detection
error.



We identify 537 derechos between 2004 and 2021 using the above objective detection criteria, with
an example of the June 2012 North American derecho shown in Figure 7. Figure 7a displays the CNN-
identified bow echoes, and Figure 7b shows the DF area and associated gust speeds. As expected, the
derecho produced extensive strong gusts during its DF period.
Although we have considered the segmentation CNN bow echo identification uncertainties in the
above derecho definition criteria, there is no guarantee that every specific situation is considered.
Therefore, we carefully examine all the identified derechos and remove 32 events that are possibly false
detections primarily due to the false identification of bow echoes (Figure S3). In addition, we manually
examine all MCS events (808 in total, excluding the aforementioned 537 automatically identified
derechos) that produce extensive strong ($\geq$ 10 observational sites) and damaging ($\geq$ 1) gusts over land
areas with a strong gust swath of at least $650 \times 100$ km$^2$. Our manual examination focuses on bow echo
identification errors but does not change any of the above derecho definition thresholds or parameters. For
those MCSs (55 events in total) that are potential derechos based on our visual inspection, we manually
label their bow echo occurrences that fail the segmentation identification during potential DF periods
(Figure S4) and rerun the automated derecho detection algorithm. Finally, 51 events meet the derecho
detection criteria described above.

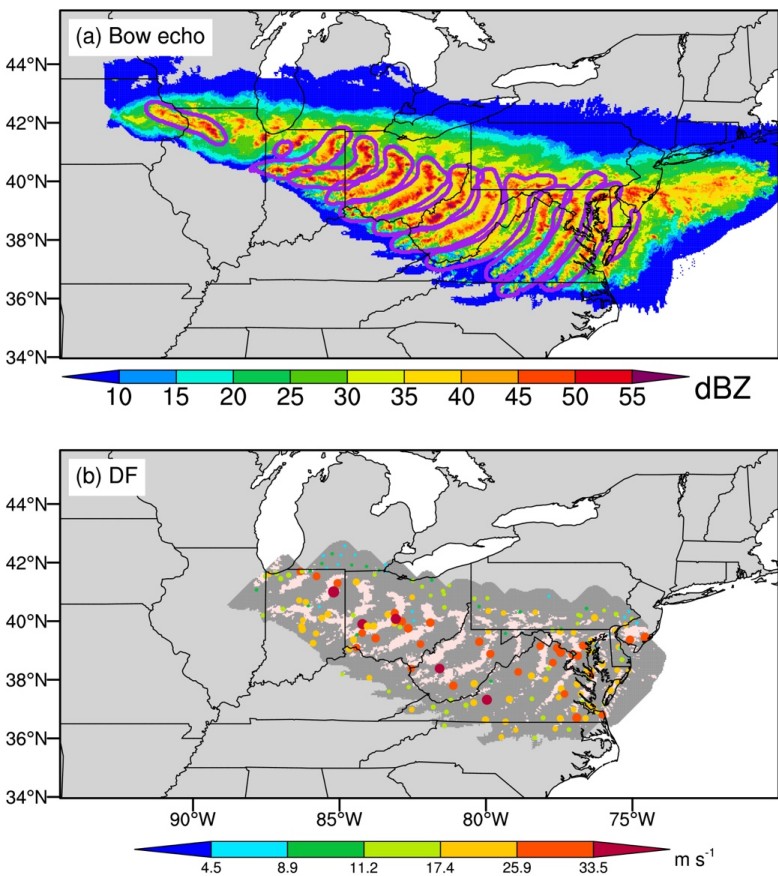

Figure 7. (a) Spatial evolution of $Z_{Hmax}$ (color shading) and CNN-identified bow echoes (purple contours) from the June 2012 North American derecho. (b) Spatial evolution of the corresponding DF. The DF lasted from 17:00 UTC on 29 June to 6:00 UTC on 30 June 2012. The misty rose shading in (b) corresponds to $Z_{Hmax} \geq 40$ dBZ, while the gray shading refers to the DF area. Colored dots are the same as those in Figure 1c, except only the DF-associated gust measurements are shown.

## 5 Dataset evaluation and uncertainty

Finally, we obtain 556 derechos between 2004 and 2021, 505 of which are identified automatically and 51 of which are added manually. The number of derechos (30.9 per year) is much larger than previous estimations (6.1-20.9 per year) using a major axis length threshold of 400 km (Squitieri et al., 2023; Johns and Hirt, 1987; Bentley and Mote, 1998; Evans and Doswell, 2001; Guastini and Bosart, 2016; Ashley and Mote, 2005). The number is also much larger than the result of Corfidi et al. (2016), which identified only 25 derechos in the warm seasons during 2010-2014 using a major axis length



threshold of 650 km. The large discrepancies are likely related to our usage of strong gusts but not
damaging gusts to calculate wind damage swath and other definition criteria. However, the diverse
observational datasets used in the derecho detections also play a critical role. Previous studies did not
have an available MCS dataset; as a result, many of their definition criteria were intended to capture MCS
events. In contrast, we have developed a high-quality, high-resolution MCS tracking dataset using
PyFLEXTKR and many coincident ground-based and remote-sensing observations. Our definition criteria
purely focus on the derecho properties and generation mechanism. Previous studies may underestimate
the derecho number due to missing MCS events. We confirm this by comparing the derechos from the
NOAA SPC with our derecho dataset in 2004 and 2005 (Table 2). The NOAA SPC data
(https://www.spc.noaa.gov/misc/AbtDerechos/annualevents.htm; last access: November 17, 2023).
provide more detailed timings and locations of derechos in 2004 and 2005 than previous studies (Squitieri
et al., 2023), which is the only available dataset that we can use to evaluate our derecho dataset at the
event scale. Notably, the NOAA SPC data contains not only derechos but also convective windstorms of
near-derecho size, and we do not know which event is a derecho or a convective windstorm of near-
derecho size. In addition, the data is based on gust speed measurements and post-disaster estimations.
There is not an underlying MCS dataset for the NOAA SPC data.
The NOAA SPC data contains 50 derechos and convective windstorms of near-derecho size, 22 of
which are directly captured by the automated detection procedure, and 2 of which can be captured after
we manually correct the segmentation CNN bow echo identification errors. Five of the 50 events are
entirely missed in the MCS dataset, possibly because they move too fast and do not meet the
PyFLEXTRKR > 50% areal overlap tracking criterion using the hourly combined satellite and NEXRAD
dataset, or they break other MCS requirements in PyFLEXTRKR (Feng et al., 2019). We emphasize that
10 of the 50 NOAA SPC events are not derechos based on the actual gust speed measurements since we
do not find any land damaging gust reports associated with the MCS events. Seven of the 50 events are
not derechos using the major axis length threshold of 650 km and the minor axis length threshold of 100
km, even if we consider all the observational sites associated with the events, regardless of whether they
are in proximity with the bow echoes. One event is an extratropical cyclone. These 18 events are excluded
from derechos using more objective or consistent criteria as NOAA SPC. The remaining three of the 50
events are missed in our derecho dataset due to the criteria used in our derecho definition, two of which
are due to too few sites with strong gusts, and one is due to the violation of the bow echo and gust speed
criteria. In summary, after excluding those 18 non-derechos and the five missed events in the MCS
dataset, the identification accuracy of our automated detection approach is $\frac{22}{50-18-5} = 81\%$ (Table 2).
Even if we consider the five missed MCS events, the accuracy can reach up to $\frac{22}{50-18} = 69\%$. For the final
derecho dataset with the 51 manually added events, the accuracy is $\frac{22+2}{50-18} = 75\%$. Finally, our derecho
dataset identifies 14 derechos that are entirely missed by NOAA SPC, confirming the underestimation of
derecho numbers in previous studies due to the lack of a reliable MCS dataset (Squitieri et al., 2023).

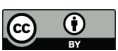

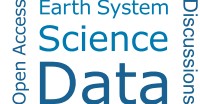


**Table 2. Evaluation of our derecho dataset against the NOAA SPC data in 2004 and 2005**

| | Year 2004 | Year 2005 | Sum |
|---|---|---|---|
| NOAA SPC[1] | 24 | 26 | 50 |
| Captured by our dataset | 10 | 12 | 22 |
| Events missed in the MCS dataset[2] | 2 | 3 | 5 |
| No land damaging gust, strong gust swath too small, no bow echo, or extratropical cyclone[3] | 10 | 8 | 18 |
| Bow echo identification error[4] | 1 | 1 | 2 |
| Other criteria not satisfied[5] | 1 | 2 | 3 |
| Our identified derechos not listed by NOAA SPC | 5 | 9 | 14 |
| Identification accuracy if excluding those missed MCS events[6] | 83% | 80% | 81% |
| Identification accuracy if including those missed MCS events[7] | 71% | 67% | 69% |

[1]NOAA SPC provides the tracks of derechos and other convective systems of near derecho size in 2004 and 2005.
[2]Some events were moving so fast that the PyFLEXTRKR algorithm, which tracks storms with spatial overlapping > 50%, could not track the systems with the hourly combined satellite and NEXRAD dataset, while some may not meet other MCS criteria.
[3]Here, "strong gust swath too small" refers to those MCS events with the largest strong gust swath of less than 650 km × 100 km, even if we include those strong gusts not associated with bow echoes.
[4]It refers to those MCS events with bow echoes not captured by the segmentation CNN. If we manually label the missed bow echoes, they would be identified as derechos.
[5]It refers to MCS events that do not meet any other criteria (e.g., too few sites with strong gusts) and cannot be classified as derechos.

$$^{6}Accuracy = \frac{Captured\ by\ our\ dataset}{Captured\ by\ our\ dataset + Bow\ echo\ identification\ error + Other\ criteria\ not\ satisfied}$$

$$^{7}Accuracy = \frac{Captured\ by\ our\ dataset}{Captured\ by\ our\ dataset + Bow\ echo\ identification\ error + Other\ criteria\ not\ satisfied + MCS\ events\ missed\ i}\quad the\ MCS\ dataset$$


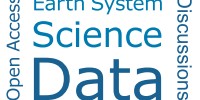

Although the evaluation against the NOAA SPC data indicates the high quality of our derecho
dataset, we must acknowledge its uncertainties caused by several sources.
The first uncertainty source is from the MCS dataset, as mentioned in the evaluation against the
NOAA SPC data. The areal overlap threshold, which is set to 50% and used to connect consecutive CCSs
in the current PyFLEXTRKR configuration, cannot capture those very fast-moving convective systems
with the hourly satellite and NEXRAD datasets. Reducing the threshold will undoubtedly increase the
"MCS" and then the "derecho" number, but it may also increase the number of false tracks that do not
belong to the same type of storm. The threshold of 50% is widely used in the different versions of the
FLEXTRKR algorithms (Li et al., 2021; Feng et al., 2023; Feng et al., 2019) and other tracking
algorithms based on overlap (e.g., (Whitehall et al., 2015)). Therefore, we would like to keep the overlap
threshold as is, but users should realize the uncertainties of the MCS dataset caused by many adjustable
parameters (e.g., area overlap threshold, MCS duration, and major axis length) and the limitations of the
observational datasets used in PyFLEXTRKR (Feng et al., 2019; Li et al., 2021).
The second uncertainty source is related to the segmentation CNN identification of bow echoes.
Although the evaluation in Section 3.3 shows the high accuracy of our bow echo identification and we
consider the bow echo identification uncertainties in the automated derecho detection procedure, we still
miss a small fraction of derechos and falsely classify some non-derechos as derechos due to the bow echo
identification error. To alleviate the CNN identification errors, we spend much effort manually examining
the derecho events identified by the automated algorithm and other MCS events that produce widespread
strong gusts. However, the manual examination is susceptible to subjective biases, and it is difficult to
completely eliminate the bow echo identification uncertainties.
The third uncertainty source is from the gust speed measurements. Although we only use gust
measurements passing the ISD quality control, it is not guaranteed that all gust speeds are reliable and
have the same quality, such as the site we exclude in Section 2.2 due to its unrealistic number of





damaging gust reports. Moreover, we cannot qualitatively evaluate the impact of the gust measurement
uncertainty on the derecho dataset, but users should be aware of the limitations of the gust speed
observations.

The last uncertainty source is related to the derecho definition criteria. Many adjustable parameters

and procedures are used in our algorithm to detect derechos. There is no doubt that changing these
parameters will alter the identified derecho number. For example, if we change the major axis length
threshold of the strong gust swath to 400 km, the derecho number will increase to 654 (a 29.5% increase).
As the first climatological derecho dataset that utilizes bow echoes in the derecho identification and
provides detailed tracking for each event, evaluating the uncertainties of the tunable parameters is
unfeasible and not our priority either. However, based on our sensitivity tests, the derecho spatial
distribution and seasonal variation patterns in Section 6 generally stay mostly the same with different
parameters (e.g., reducing the strong gust fraction threshold to 10% or the threshold of the number of sites
with strong gust reports to 5). The exception is that when we calculate the gust swath length and width
using sites (requiring ≥ 10 sites) with damaging gusts as in previous studies (Squitieri et al., 2023), the
derecho number is significantly reduced to 19, highlighting the spatial limitation of ISD gust
measurements. We emphasize that although our derecho definition follows Corfidi et al. (2016), we
exclude the "forward propagating" criterion they proposed. We try several methods to calculate the angles
between the derecho orientations and their propagation directions but cannot obtain satisfying and
accurate results for some events with complex structures. Figure 8 shows the probability density function
(PDF) of the angles between "derecho propagation directions" and "bow echo orientations" for all
derechos from the final derecho dataset. Based on this type of calculation, 78% of derechos have an angle
≥ 30°, and 58% of derechos have an angle ≥ 45°. For those derechos with angles < 30°, it does not mean
that they are not forward propagating systems, but it is more likely that this type of angle calculation does
not reflect their actual propagation direction. In total, even though we do not use the "forward
propagating" criterion in the derecho definition, most of the identified derechos are indeed forward
propagating systems.

Finally, users should acknowledge the high quality of our derecho dataset but understand its

limitations due to various uncertainties during its generation.

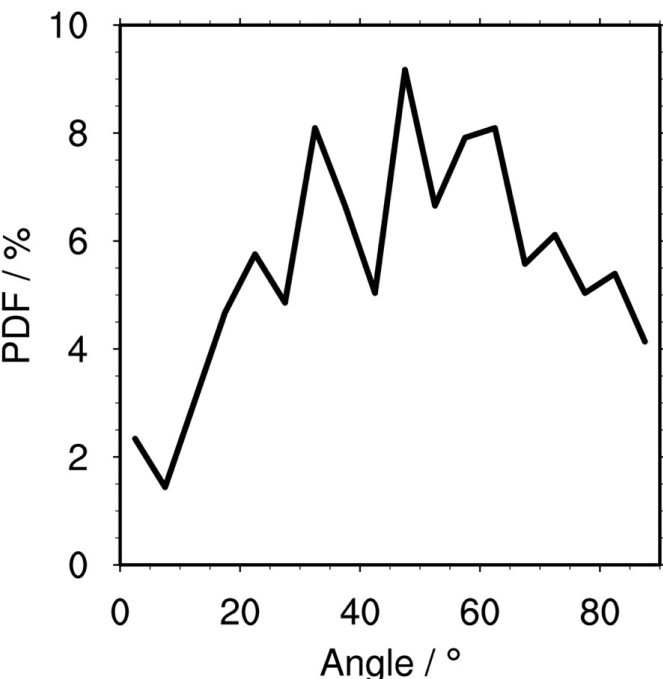

Figure 8. The probability density function (PDF) of the angles between derecho propagation directions and
bow echo orientations. For any derecho, we calculate all the bow echoes' orientations during its DF period and
use the median orientation in the angle calculation. Propagation direction is also based on bow echoes during
the DF period. We select any two distinct bow echoes during the period and use their centroid points to derive
a direction. If there are n bow echoes, we can obtain $C_n^2 = \frac{n \times (n-1)}{2}$ directions. Similarly, we use the median
direction as the derecho's propagation direction to calculate the angle. The angle is initially in the range of -
180° to 180°, and we adjust them to be between 0° and 90° to reflect the minimum angle between the
derecho's orientation and propagation direction.

## 6 Derecho climatological characteristics

We use the final derecho dataset with 556 derechos to conduct the following climatological analyses.





## 6.1 Annual statistics


Figure 9 displays the annual derecho numbers from 2004 to 2021. There is an apparent jump in the
derecho number before (~20 derechos per year) and after 2007 (~30 derechos per year), which may be
partially related to the general increase in the number of gust speed observational sites from 2004 to 2010
(Figure S5). Figure 10 shows the spatial distribution of yearly averaged annual derecho numbers between
2004 and 2021, and the derecho paths during their DF periods are displayed in Figure S6. The central
Great Plains has the most frequent derecho occurrences, extending to Oklahoma in the south, Iowa in the
north, Kansas in the west, and Illinois in the east. The areas with frequent derecho occurrences are
generally consistent with previous studies (Coniglio and Stensrud, 2004; Guastini and Bosart, 2016; Johns
and Hirt, 1987; Ashley and Mote, 2005), although some differences are identified. For example, several
studies identified a northwest-southeast axis with frequent derecho occurrences extending from southern
Minnesota to Ohio, which is not apparent in our spatial distribution (Johns and Hirt, 1987; Coniglio and
Stensrud, 2004; Guastini and Bosart, 2016). The differences can be caused by many factors, such as
distinct derecho definitions and observational datasets used in these studies. We make a sensitivity test by
calculating the gust swath using ≥ 10 sites with damaging gusts as mentioned in Section 5, which
identifies 19 derechos. The corresponding spatial distribution in Figure S7 well captures the
aforementioned west-east axis, although the occurrence frequency is much smaller than in previous
studies with more than one derecho occurrence per year (Johns and Hirt, 1987; Coniglio and Stensrud,
2004; Guastini and Bosart, 2016). The sensitivity test seems to indicate that the most intense derechos
prefer to occur in the northern Great Plains and Midwest, while weaker derechos occur preferably in
central Great Plains around the junction of Oklahoma, Kansas, Missouri, and Arkansas.

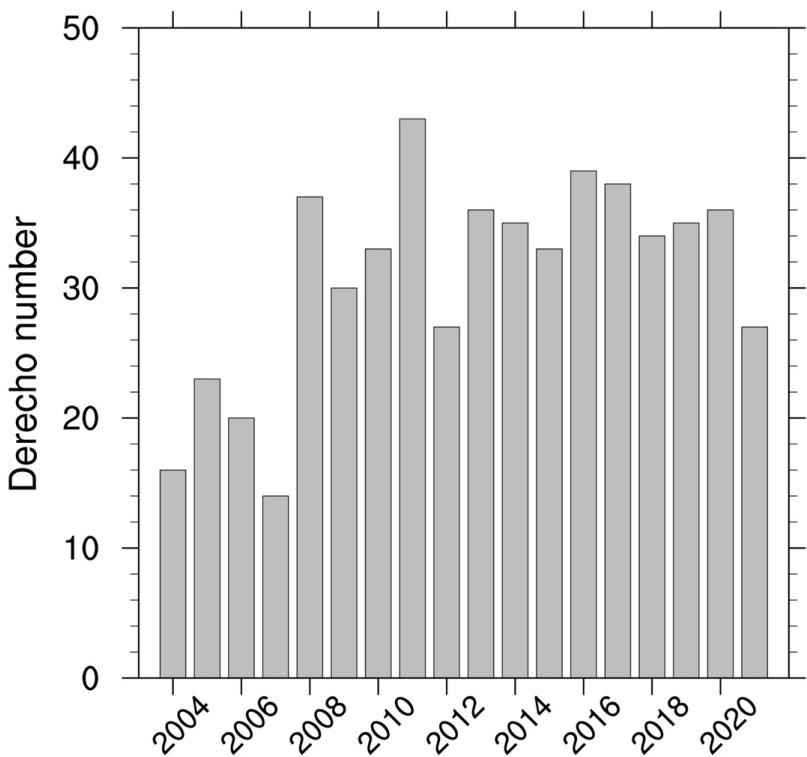

Figure 9. Bar chart of the annual derecho numbers from 2004 to 2021.

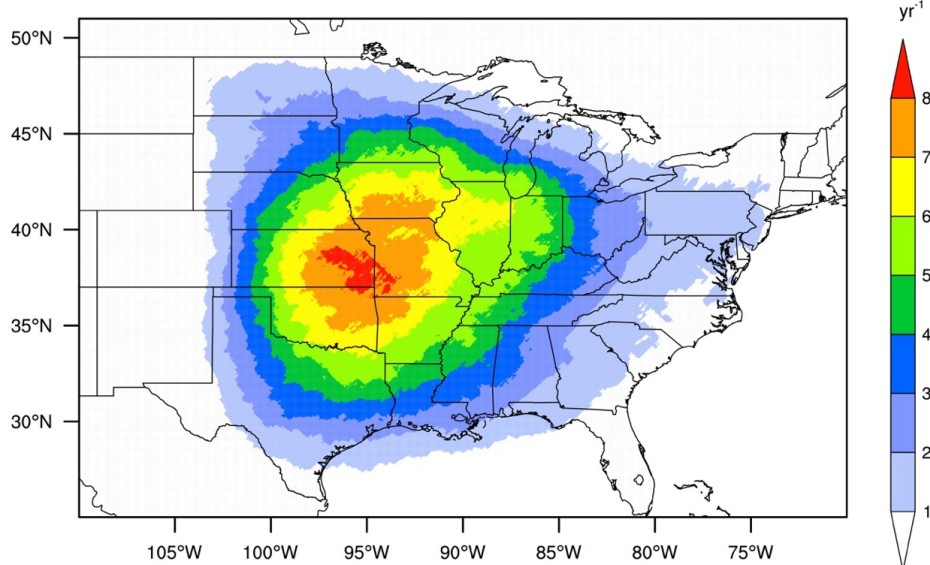

Figure 10. Spatial distribution of yearly averaged annual derecho numbers over the United States east of the
Rocky Mountains between 2004 and 2021. Here, we use derecho DF areas as derecho spatial coverages.



## 6.2 Monthly statistics


Figure 11 displays the yearly averaged seasonal variations in the derecho number, with remarkably

more derechos in the warm than cold seasons, a feature widely captured by previous studies (Ashley and

Mote, 2005; Squitieri et al., 2023; Bentley and Sparks, 2003). The derecho seasonal variation resembles

that of the MCS events (Feng et al., 2019), similar to the derecho annual spatial distribution (Figure 10

and Feng et al. (2019)).

Figure 12 shows the spatial distributions of the yearly averaged monthly derecho numbers between

2004 and 2021. On the one hand, many more derechos occur in the warm than cold months. On the other

hand, we find remarkable shifts in the areas with the most frequent derecho occurrences from April to

August. The region with the most derechos moves northward during the warm season but shrinks zonally.

The northward shifts also resemble the MCS events (Li et al., 2021). We can identify two axes with

frequent derecho occurrences. One is in the south-north direction along the Great Plains, and the other is

in the west-east direction along the northern Great Plains and Midwest, which are consistent with the

derecho paths in Figure S6. The axes may represent the two types (serial and progressive) of derechos

mentioned in Squitieri et al. (2023). A follow-up study will be conducted to investigate the large-scale

environmental conditions associated with different types of derechos based on the developed derecho

dataset. Notably, derechos are concentrated in the Lower Mississippi Valley in the cold season, which is

also consistent with previous studies (Squitieri et al., 2023).

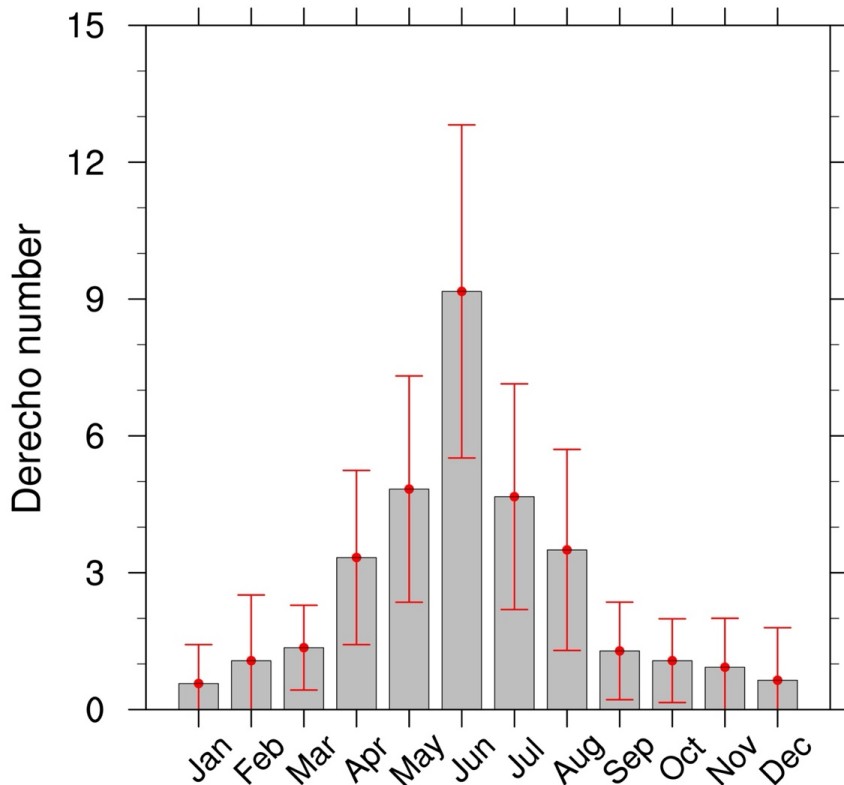

Figure 11. Yearly averaged monthly variations of the derecho numbers between 2004 and 2021. The error bars
denote standard deviations.

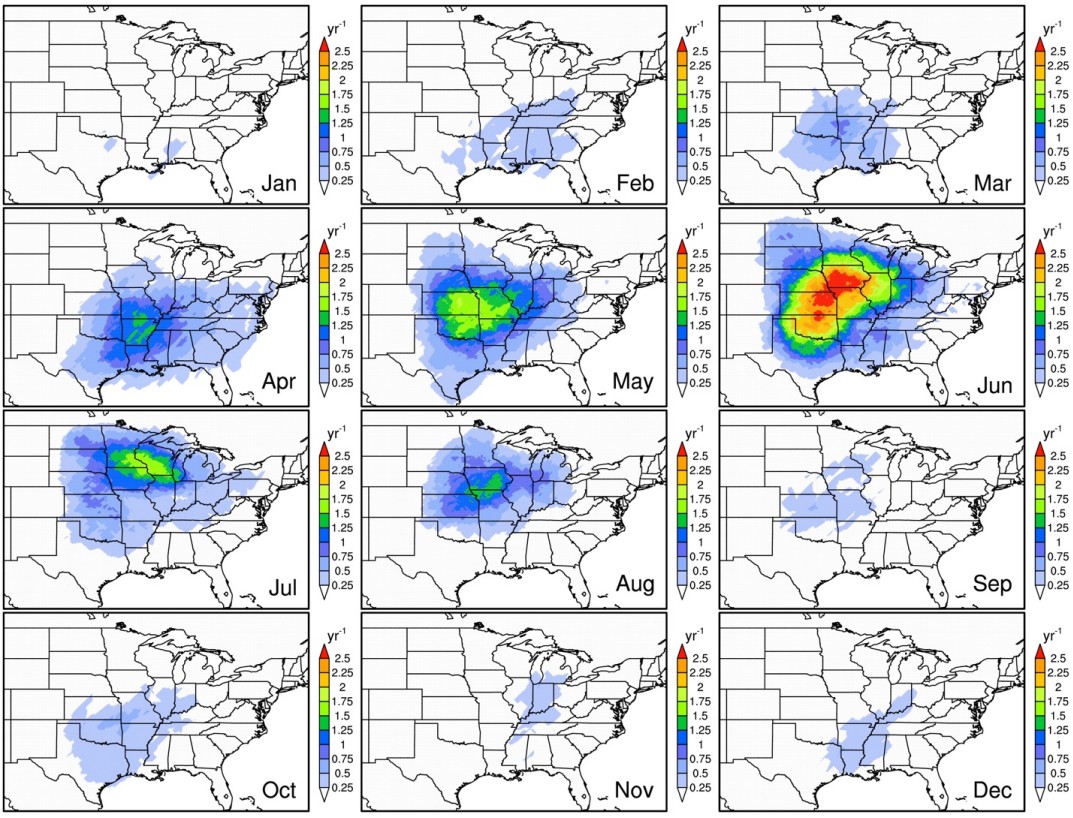

Figure 12. Same as Figure 10 but for yearly averaged monthly derecho numbers over 2014-2021.
**6.3 Wind damage characteristics**
We examine the contributions of derechos and DFs to all the damaging gust reports in the United
States area of the dataset domain between 2004 and 2021 in Figures 13, S2, and S8. MCSs contribute
about 36.8% of the damaging gust reports, but most occur east of the Rocky Mountains. On average,
derechos and DFs contribute 19.2% and 16.5% of the damaging gust occurrences, respectively. In other
words, about half of the damaging gusts associated with MCS events are related to derechos.
Understanding the underlying mechanisms will be our focus in a follow-up study. In addition, most (>
80%) derecho-generated damaging gusts occur during the DF periods, justifying using DF in our derecho
definition, consistent with the larger probabilities of extreme gusts in the gust speed PDF of DFs than that



of derechos in Figure S9. The gust speed PDFs for MCSs and derechos indicate that derechos are more
favorable for producing extreme gusts than MCSs (Figure S9). Moreover, as expected, the contributions
of derechos to damaging gust reports are the highest in the Great Plains, Midwest, and Lower Mississippi
Valley (Figure 13).

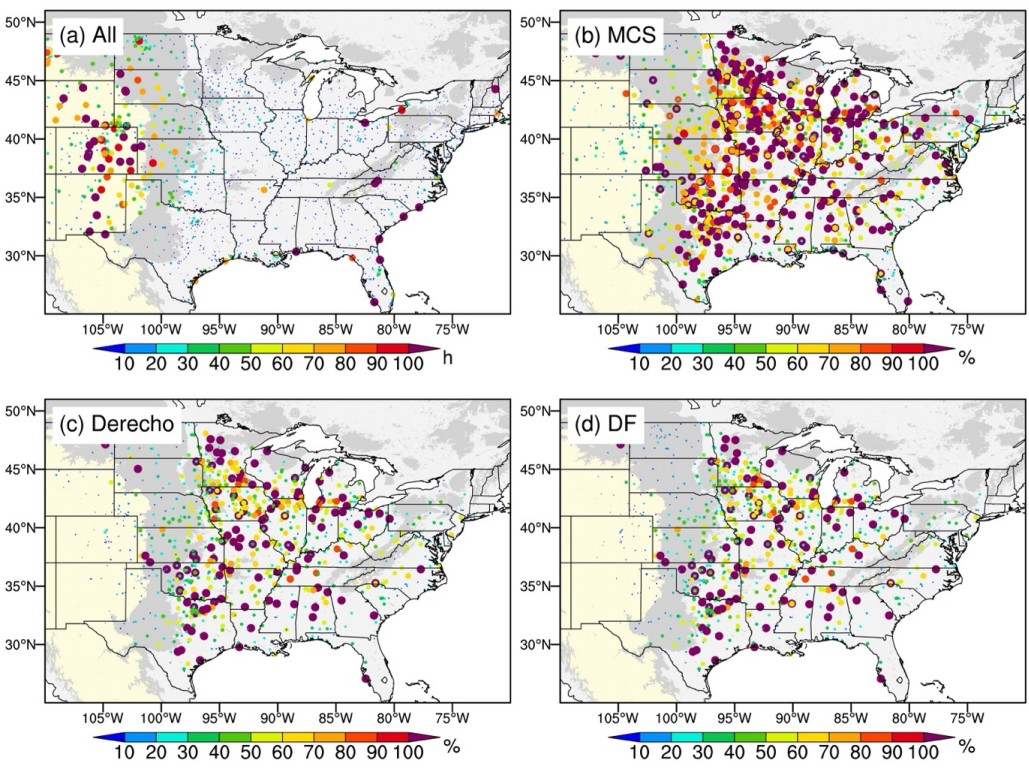

Figure 13. (a) The total numbers of damaging gust occurrences between 2004 and 2021 at weather stations
over the United States east of the Rocky Mountains. (b) Relative contributions of MCS events to the damaging
gust occurrences in (a). (c) is the same as (b) but for relative contributions of derechos. (d) is the same as (c),
but we only consider the DF periods when counting the derecho-associated damaging gust occurrences.
Similar to Figure 5, we exclude non-derecho MCS events overlapping with TCs in (b). The dot sizes are
proportional to the corresponding values. Light-yellow shading denotes an elevation greater than 1000 m;
light-gray shading denotes an elevation between 400 m and 1000 m; and smoke-white shading denotes an
elevation less than 400 m. Background white is for oceans and lakes.





## 7 Data availability


The final derecho dataset and the corresponding user guide are available
at https://doi.org/10.5281/zenodo.10884046 (Li et al., 2024). The original format of the data files is
NetCDF-4, and we compress them for each year so that the dataset is easily accessible. The user guide
contains a detailed description of the data files to help users understand the dataset. For each derecho, the
dataset provides two figures displaying the temporal evolutions of $Z_{Hmax}$, precipitation, wind speed, and
gust speed during its entire lifetime and DF period (e.g., Figures 14 and S10). The figures are helpful for
users to understand the basic characteristics of the derechos immediately. Notably, the dataset contains all
the derecho-associated gust speed measurements, so users can further separate the derechos into different
intensities, as in Coniglio and Stensrud (2004).

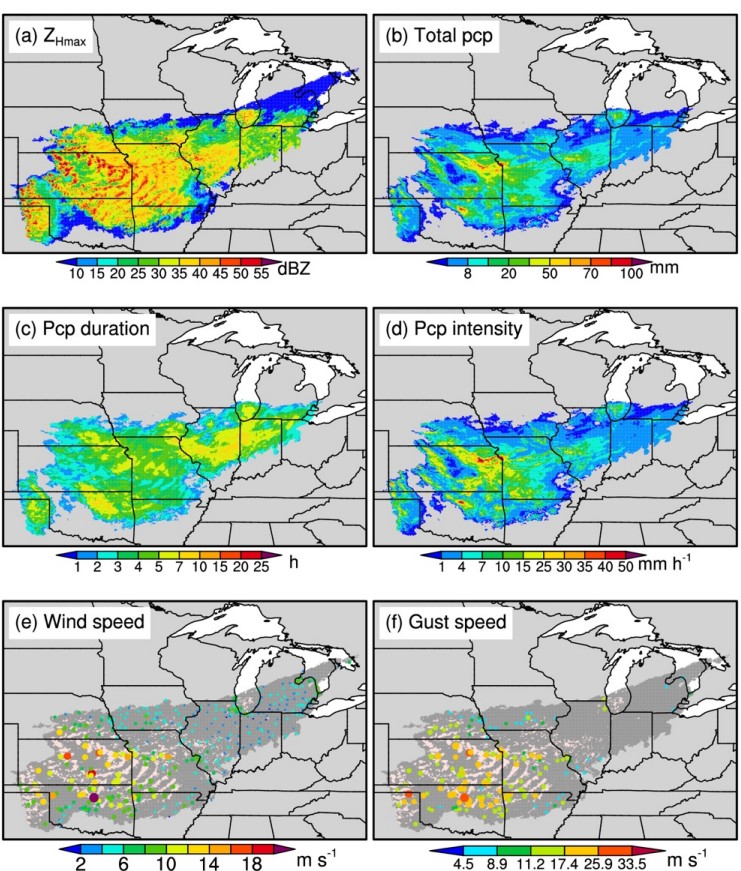

Figure 14. Similar to Figure 1 but for the spatial evolutions of (a) $Z_{Hmax}$, (b) total accumulated precipitation, (c)
precipitation duration, (d) mean precipitation intensity, (e) hourly maximum wind speed, and (f) hourly
maximum gust speed during the entire lifetime of a derecho that occurred on 10-11 September 2015. In (e) and
(f), the misty rose shading corresponds to areas with $Z_{Hmax} \geq 40$ dBZ, and the dark gray shading refers to
derecho coverage with $Z_{Hmax} < 40$ dBZ. The figure title refers to the derecho timing range.

## 628    8 Conclusions

This study presents a high-resolution (4 km and hourly) observational derecho dataset covering the

United States east of the Rocky Mountains from 2004 to 2021. We develop the dataset using an MCS
dataset generated by the PyFLEXTRKR software, a machine-learning-based identification of bow echoes,
ISD hourly gust speed measurements, and physically based identification criteria. The evaluation and
potential uncertainties of the dataset are discussed. The dataset contains 556 derechos, most of which are



in the warm season (April-August). Analyses indicate that derechos preferably occur in the Great Plains
and Midwest. Areas with the most frequent derechos show a northeastward shift from April to August.
Derechos contribute 19.2% of land damaging gusts over the United States between 2004 and 2021. About
half of MCS-associated damaging gusts are produced by derechos. As the first derecho dataset that uses
machine-learning identification of bow echoes, physically based definition criteria, and surface station
measured gust speeds, it provides an independent reference for derecho climatology compared to previous
studies. In addition, the derecho dataset can be used to investigate the derecho initiation and development
mechanisms, the environments that facilitate the formation and intensification of derechos, and the
damage of derechos to human security and property. Moreover, due to its high spatiotemporal resolutions,
the dataset can be used to select specific derecho events for case studies and evaluate the numerical model
simulations.

## Author Contributions

JL, ZF, and LRL designed the study. JL prepared the input files for PyFLEXTRKR, and ZF ran
PyFLEXTRKR to generate the MCS dataset. JL and ZF generated the initial positive and negative bow
echo samples. AG trained and validated the CNN model. AG applied the trained semantic segmentation
CNN to identify bow echoes from the MCS dataset with discussions with JL and ZF. JL defined and
identified derechos with discussions with ZF. JL evaluated the derecho dataset and manually examined
the data. JL analyzed the derecho climatology with discussions with ZF. JL wrote the manuscript except
for the machine-learning part which was written by AG. All co-authors reviewed the manuscript.

## Competing Interests

The authors declare that they have no conflict of interest.



## Acknowledgments


The NOAA SPC derechos and near-derechos are available at
https://www.spc.noaa.gov/misc/AbtDerechos/annualevents.htm (last access: November 17, 2023). The
named derechos we use to generate bow echo samples are from
https://en.wikipedia.org/wiki/List_of_derecho_events (last access: 19 March 2023). The elevation data is
from http://iridl.ldeo.columbia.edu/SOURCES/.NOAA/.NGDC/.GLOBE/ (last access: March 7, 2024).
The IBTrACS Version 4 TC data over the North Atlantic basin is from https://doi.org/10.25921/82ty-
9e16 (Knapp et al., 2018). Thank Drs. Israel L. Jirak, Brian J. Squitieri, and Andrew R. Wade from
NOAA SPC for discussing the derecho definition criteria with us.
The bow echo segmentation code and datasets are available at https://doi.org/10.5281/zenodo.10822721
(Geiss et al., 2024). This repository includes the trained CNN weights and instructions for use. A video
supplement showing the bow echo segmentation scheme in use can be viewed at
https://youtu.be/iHWY_OhaVUo and is permanently archived in the above Zenodo repository.

## Financial support


This research is supported by the Regional and Global Model Analysis and Multisector Dynamics
program areas of the U.S. Department of Energy Office of Science Biological and Environmental
Research as part of the HyperFACETS project. PNNL is operated for the Department of Energy by
Battelle Memorial Institute under Contract DE-AC05-76RL01830.



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
