# Peer review of "A derecho climatology (2004-2021) in the United States"

_Earth System Science Data, 2024_

## Author Comment (AC1)

**Response to Reviewer #1**

Thank you for your careful and thorough reading of the manuscript and your thoughtful comments and suggestions. We apologize for the delay in revising the manuscript, as we spent a significant amount of time on the manual validation and inspection of the identified derechos. According to all three reviewers' comments and suggestions, we have made several significant improvements in our manuscript, which we want to highlight before the point-by-point response.

Firstly, we have changed the wording of some terms so that they are more distinguishable and accurate, and we have also improved the language of many sentences so that the manuscript is more understandable. A derecho is a windstorm, while an MCS is a convective system. They are different concepts. Therefore, we change "derecho" to "derecho-producing MCS" when we refer to the MCS associated with a derecho, and "derecho feature" has been changed to "derecho." The dataset developed in this study includes tracking of both derechos and corresponding derecho-producing MCSs.

Secondly, we have incorporated a "forward propagating" criterion in our derecho detection algorithm. Our initial understanding of the term "forward propagating" was incomplete, and we failed to recognize its critical role in defining a derecho proposed by Corfidi et al. (2016). After careful consideration and evaluation, we have adopted and modified two criteria from Corfidi et al. (2016) to establish the definition of "forward propagating." One is that the acute angle between the averaged bow echo orientation and the bow echo series' propagation direction is larger than 45°, and the other is that the bow echo series' propagation speed is at least 30% faster than the 500-hPa background wind speed. Implementing the "forward propagating" criterion removes many windstorms externally forced by extratropical cyclones, aligning with the purpose of Corfidi et al. (2016), which intends to define derechos as internally driven windstorms. "Externally forced" and "internally driven" reflect distinct physical formation mechanisms of those windstorms, which is why Corfidi et al. (2016) proposed a physically based derecho definition. With the updated detection algorithm, the derecho number between 2004 and 2021 has been substantially reduced from 556 to 274 (for ISD) and 220 (for SED). In addition, due to the inclusion of the "forward propagating" constraint in our derecho algorithm, we have decided not to change the name of our dataset to "high wind-producing bow echo." We have updated all the results in Section 6 based on the improved dataset.

Thirdly, we have developed another parallel dataset using gust speeds from the Storm Events Database (SED). Now, our derecho dataset consists of two subsets: one based on gust measurements from the global hourly Integrated Surface Database (ISD) and the other based on SED gust speeds. Although there are some discrepancies between the two subsets, their agreement is much larger than their difference (Figures 9-12 in the revised main manuscript; or Figure R1 below). Moreover, both ISD and SED gust speeds have limitations and uncertainties, hence differences between the ISD-based and the SED-based datasets are expected and understandable. In addition, it also indicates that our usage of lower gust speed criteria for ISD measurements is reasonable and does not change the derecho number much. We must emphasize that using lower gust speed criteria for ISD measurements than SED reports does not mean that the ISD-identified derechos are weaker than the SED-identified ones (or even not derechos). This

is a compromise, considering that ISD stations are limited and may miss many damaging gusts, as we highlighted in Lines 460-469 in the revised main manuscript (as below).

"We emphasize that, in Criterion 4, our ISD gust speed criteria are weaker than the SED gust speed criteria as well as those of previous studies (Squitieri et al., 2023; Bentley and Mote, 1998; Johns and Hirt, 1987), which also estimated the gust swath based on SED damaging gusts. As mentioned in Section 2.2.2, most SED gust reports are estimates, while ISD provides gust measurements from weather stations. SED estimates can capture potential damaging gust occurrences over a much larger area, although with large uncertainties. In contrast, due to the limited coverage of observational sites, real-time ISD measurements may miss substantial damaging gust occurrences in nearby regions. Therefore, we lower the gust speed criteria to capture potential derechos when using ISD measurements. It does not mean that the ISD-based derechos are weaker than the SED-based ones or even not derechos, as elaborated in Section 5."

Fourthly, due to the incorporation of "forward propagating" in the detection algorithm and the development of the SED-based derecho dataset, we have updated some sensitivity tests, evaluations, and the comparisons of our datasets with the NOAA SPC data in 2004 and 2005 and previous studies. Please see Section 5 for further details.

[Figure]

Figure R1. Bar chart of the annual derecho numbers from the ISD-based and the SED-based datasets from 2004 to 2021. Gray shading denotes derechos captured by both datasets, red shading refers to derechos only identified when using ISD gust observations, and blue shading represents SED-only derechos. The figure is the same as Figure 9 in the revised main manuscript.

*The general idea of an AI-based objective tool to identify derechos is fascinating, and such a tool could be very useful. The approach taken in this paper has promise and it is exciting that some reasonably good results are obtained. However, I feel like there were some poor choices made in the design of the tool that harm its performance, and I list those below. In addition, I wonder if the timing of a paper like this is not good, in light of the fact that a very new physics-alone-based definition of a derecho is being proposed by some severe weather scientists (e.g. Squitieri 2024, abstract 16A.3 for AMS 28th Conf. on Sev. Local. Storms). In my own work with derechos and severe thunderstorm winds in recent years, I was aware that many in the community feel the definition much change from its current unusual focus on after-the-fact wind measurements. In meteorology, we normally define things based on their physics, and it sounds like that is what is being proposed by the severe weather community within SPC, which typically has been tasked with identifying derechos. It seems it would be much better for the AI work in this present study*

*to be better coordinated with the group proposing the new definition, so that the AI tool is designed to work with the new definition. As it stands, I fear the AI tool will be obsolete from day 1, particularly in light of my concerns listed below. The Squitieri 2024 abstract indicates that the new definition reduces the number of derechos per year substantially, which would make the present study's overestimate of derechos even worse. In summary, I believe the problems associated with the design of this tool are so severe that the paper cannot be published in its current form. The AI tool is really just identifying bow echoes that produce strong winds, not derechos. This presents numerous fatal problems in the discussion of results.*

Reply:

Thank you for your comments and interest in the AI tool. As discussed in the manuscript, the segmentation CNN is used to identify bow echoes, which is only part of the derecho detection. We shared the bow echo dataset as well as the segmentation CNN code with the public on Zenodo. Any group can use the AI tool or the CNN-identified bow echoes in their studies, regardless of whether they are related to derecho detection or not.

Before submitting the manuscript, we discussed the derecho definition issue with the group that submitted the AMS abstract you mentioned. Unfortunately, we did not reach an agreement on the derecho definition. According to our understanding, except for distinct approaches to identifying bow echoes, the primary difference between our algorithm and their derecho definition is the gust speed criteria. The AMS abstract used gust speeds from SED and required a derecho to have at least five reports with gust speed $\geq 33$ m s$^{-1}$ in the 400-km long wind swath, with at least three of the $\geq 33$ m s$^{-1}$ gust reports measured; otherwise, there must be at least three damaged-estimated gust reports with a speed $\geq 44$ m s$^{-1}$. We understand their motivation to use much stricter gust speed criteria. Those derechos have more significant socioeconomic impacts than relatively weaker ones. But it's arguable that their derecho definition is physics-based, while ours is not. Both definitions are based on Corfidi et al. (2016), where the physical parts are more reflected in the usage of bow echoes with mesovortex and/or rear-inflow jets, cold-pool driven, and forward propagating. We do not mean that different gust speeds are all generated by the same physical mechanism. They may be associated with different physical processes (which requires additional analysis after we develop a derecho dataset), but in the definition, the gust speed criteria are more related to socioeconomic impacts. As we mentioned above, even if we use SED gust reports and the same gust speed criterion from Corfidi et al. (2016) in our derecho detection algorithm, the derecho number is still similar to that based on ISD with a weaker gust speed criterion. We used a lower gust speed criterion for ISD because of the limitations and uncertainties of the gust speed dataset, and it does not mean that the ISD-based derechos are weaker or that they are not derechos. Moreover, even if NOAA plans to use stricter gust speed criteria in the future, it does not undermine our study. Our dataset is flexible, and we store all the key parameters for each derecho in the dataset, including gust speed reports from SED or ISD. Users can easily filter out stronger derechos using existing variables in our dataset if they are interested in the strong derechos, which would reduce the number of events in the full derecho dataset.

We also want to clarify that we do not intend to use the algorithm for operational purposes by developing the detection algorithm and derecho dataset. The segmentation CNN bow echo

identification is imperfect and could suffer from uncertainties, which is why we invested substantial time on manual inspection of the dataset and those MCSs that produced strong gusts. For meteorological operations in the United States, instead of manually correcting the AI detection errors, it may be easier for an experienced meteorologist to label bow echoes and identify derechos manually when the data volume is small. When the data volume is large, as in this application with 18-years of hourly data, the AI classifier is much more efficient. Besides developing the 18-year derecho climatology over the United States, we expect that the CNN bow echo identification tool and the automatic derecho detection algorithm can be used in other areas including application to model results with large volumes of data. This is another reason why we prefer to use the ISD-based derecho subset instead of the SED-based subset: other countries often do not have as much data as the United States, especially the SED data. However, many countries have some measurements from weather stations, and they can easily adapt our detection algorithm to fit their specific conditions.

*I am curious since prior definitions of derechos have included the length requirement (roughly 400 miles of damaging winds, or 650 km, in the more recent stricter definition of Corfidi et al. 2016 that you cite), why you would shorten the MCS longevity requirement to just 6 hours? If one assumes a typical MCS translation speed of 50 km/h, then an MCS lasting only 6 hours would only move over an area 300 km long, or less than half the pathlength needed in this prior definition of derechos. Even with the older 400 km pathlength requirement that you mention, many events would not meet it if moving at 50 km/h. Also, even if one acknowledges that derechos often move rather quickly (I believe the 2020 Midwestern one moved at around 90-100 km/h, which is about as fast as they can move), a 6-hour lifetime would still not result in a damage path quite long enough to match the 650 km definition. It is true that MCSs can produce significant damage with wind swaths shorter than 650 km, but if that is the focus of your work, you should not be referring to it as a derecho climatology. A high wind-producing bow echo climatology would be more appropriate for the title. In fact, in light of my opening paragraph and the ongoing efforts to change the definition of a derecho substantially, this change in title might avoid even more serious problems. It much better defines what your tool does since it is broader in its wording. I would strongly suggest that change.*

Reply:

Thank you for your comments. The wording in the last version of our manuscript might cause some confusion. We have improved the wording of some terms, as explained at the beginning of the response, which we hope can address your concern. In detail, the study aims to identify derechos (and derecho-producing MCSs) from the MCS dataset, which comprises derecho-producing MCSs and non-derecho-producing MCSs. Using a lifetime threshold of 6 hours in our MCS detection algorithm does not affect our derecho (and derecho-producing MCS) detection, which explicitly requires a gust swath of at least $650 \times 100$ km$^2$ associated with an MCS (Line 398 in the revised main manuscript, or as below).

"The gust swath must be at least 650 km in length and 100 km in width."

According to our current dataset, the average derecho and derecho-producing MCS lifetimes are 11.4 and 27.8 hours respectively when we use ISD in our derecho detection. The numbers change to 10.5 and 29.4 hours when SED is used in the derecho algorithm.

According to our dataset, the typical derecho propagation speed is about 25 m s$^{-1}$, hence 650 km is roughly equivalent to 7 hours. Considering the potential CNN bow echo identification uncertainties, we change the lowest derecho duration from 2 hours to 5 hours in Line 378 in the revised main manuscript. In fact, the lifetime threshold is not so important since we have set a swath threshold of $650 \times 100$ km$^2$, which already implies a duration threshold.

"The derecho must persist for at least 5 hours, with a bow echo present for at least 80% of its lifetime."

*I am troubled by your use of the surface wind station database for the winds used in your system. This is NOT how derechos are classified operationally now. Instead, classification is based on the Storm Events Database (SED), which I would assume includes more actual measurements (such as home weather stations) and estimates of wind speed based on damage. In fact, roughly 90% of all the severe wind reports in that database do not involve a measurement and are instead estimates based on damage. The definition of a derecho that I found on SPC's website does not restrict derecho classification to just measured winds. The Storm Events Database is rather robust for the period 2004-2021 that you are using, and thus it is puzzling why you would not have used that for your training? Your use of just the surface station database is even more puzzling considering my comment earlier that your reduction in the longevity requirement for MCSs would lead your system to call many events derechos that would not meet the past requirements of a swath of severe winds at least 650 km long (or even 400 km long in many situations). Thus, you made one choice that really makes it easier to call a system a derecho, but then this choice of where you obtain wind information would do the opposite, making it much more difficult for systems to meet prior thresholds to be called a derecho. Tirone et al. (2024) used the SED in their training of a ML tool, so there is no obstacle to using that information as a source of thunderstorm wind information. I believe you need to test the sensitivity of your results to your use of a very limited database of thunderstorm winds, by examining changes that happen if you switch to using the SED. I see that later starting at about line 350, you provide good explanation which I think deals with my concern. It seems you acknowledge the deficiencies in your choice of this surface data network for wind and thus make numerous modifications to try to account for the deficiencies (like lower wind thresholds, use of broader ellipse containing reports, etc). You talk some about how damage estimates take some time to be performed, but it is unclear if this is the primary motivation for your use of the problematic surface wind network dataset. I believe you need to provide some of this justification earlier when you first mention that surface network, since most readers familiar with severe weather reports will question why you are not using the SED. Derechos are often classified now within 24 hours of their occurrence as the process of gathering both the measurements and having some estimates is quick. From what I have read so far, I do not see a reason why 24 hours is too long for what you are doing. You need to make a stronger case for your use of the surface wind measurements. Are you planning for your tool to be used operationally in a setting where it must alert forecasters the moment that a system has reached the requirement to be called a derecho? I guess I do not see why this would be so urgent. With all ongoing derecho research that I am aware of, your tool could easily be applied one day after*

*an event, or even a week or month afterward, so there does not seem to be a valid reason to avoid using the SED.*

Reply:

Thank you for your comments. Please see our above answers and clarifications. Besides, we also want to clarify that our AI tool is only used to detect bow echoes from composite radar reflectivity. The training of our bow echo detection tool does not use gust speed.

*When you refer to bow echo samples on line 180, it would be helpful to know the time resolution of the radar images you use. In recent years, radar reflectivity images often update nearly every minute or two. For readers to be able to put into context your 566 positive identifications, they need to be able to figure out how many images in total might be getting evaluated in the normal lifetime of a derecho. It is possible the citation about Gridrad would mention the time frequency of the Gridrad products, but this is simple information to supply in your own paper and is absolutely necessary. For a derecho lasting 8 hours, if radar images are available every 5 minutes (the traditional frequency of NEXRAD scans), there would be 96 for one case, and thus your 566 total bow echo scans from 54 events would be a tiny fraction of the lifetime of the derechos. If the images are hourly, then 566 implies every hour of every derecho must include a bow echo. Thus, this information is critically important.*

Reply:

Thank you for your comment and suggestion. Gridrad has a temporal resolution of 1 hour. We added "hourly" before "source datasets" in Line 131 in the revised main manuscript.

'Hourly source datasets are used in the generation of the MCS dataset, including the National Centers for Environmental Prediction (NCEP)/the Climate Prediction Center (CPP) L3 4 km Global Merged IR V1 brightness temperature dataset (Janowiak et al., 2017), the three-dimensional Gridded NEXRAD Radar (GridRad) dataset (Bowman & Homeyer, 2017), the NCEP Stage IV precipitation dataset (CDIACS/EOL/NCAR/UCAR & CPC/NCEP/NWS/NOAA, 2000), and melting level heights derived from ERA5 (European Centre for Medium-Range Weather Forecasts (ECMWF) Reanalysis v5) (Hersbach et al., 2023).'

*I believe the results you state starting at about line 422 reflect some of the harm done by choices you made that I take issue with in my earlier comments. You do mention your reduction in wind threshold, but this could likely have been avoided had you used the SED instead of the surface wind network. Likewise, your overestimate compared to other studies is probably influenced by your unusual choice to reduce the MCS longevity criterion. You should try using a longer threshold than 6 hours to see how the numbers change. I do not believe you have made a case as to why you needed to reduce it to 6 hours. If your line 433 is implying that there may be derechos in the SPC database that are not really derechos since they did not come from organized convection, I highly doubt that. SPC usually avoids even showing wind damage reports unless they are clearly due to convection. There is no way I can conceive that a system would get listed as a derecho, because of that path length requirement), if it did NOT come from an MCS. Perhaps I have misinterpreted what you say in line 433, but if so, you need to be more direct in explaining why your MCS criteria is so important. From what I know about the SPC*

*database, all events being called derechos had to be MCSs. For your AI work, obviously you need to have some criteria to ensure a system is an MCS, but when humans classify derechos, they are automatically ensuring this. Thus, I think the only impact of your MCS criteria would be if you are comparing your number of cases to numbers in other studies that used AI to classify derechos (and presumably to explain why you should end up with a smaller number than other studies that may not have bothered to ensure the winds were happening due to an MCS). In the context here, in your paper, however, I do not believe it makes sense.*

Reply:

Thank you for your comment and suggestion. Since we now include the SED-based dataset in the study, we have rewritten Section 5 about the evaluation and uncertainties of our dataset. Please also see above for our answers and clarifications.

*Your discussion around line 450 again seems like it is needed there only because you did not use the SED database. If you had, you could avoid having to explain so many possible caveats.*

*Once again, starting around line 552, this discussion is reflecting the serious flaw in your design of your experiment. You chose to define derechos in a way inconsistent with the already problematic fact that at least two standard definitions exist. It is like you chose to study cats, but are calling them dogs, and now you are having to explain numerous differences between your study and prior ones. Because you used a different definition, it is impossible to know how serious the differences are. As I stated earlier, you are really just identifying bow echoes that were associated with strong winds. Not derechos. The question here becomes, is there a physical reason why strong wind bow echoes do not show the NW-SE swath of enhanced occurrence that is present for derechos? Or is there some fundamental issue with AI that is resulting in the difference. We cannot know because you chose to compare apples to oranges. The sensitivity test you do on line 554 is interesting and may offer some insight into my question above. But I believe your neglect of using the SED still complicates the interpretation you are providing here.*

Reply:

Thank you for your comment. In this study, we do not propose a new derecho definition but follow the definition from Corfidi et al. (2016). However, due to the limitation of some source datasets, we modify some thresholds, especially for those related to ISD gust speeds. This is a compromise. The derecho definition from Corfidi et al. (2016) is different from conventional ones. Since there are no other derecho climatologies based on Corfidi et al.'s definition, we can only compare our dataset with results based on conventional definitions. We disagree that this is a comparison between apples and oranges. The derecho definition is continuously evolving. Even the one in the AMS abstract you mentioned is different from previous definitions. On the other hand, the NW-SE swath of enhanced occurrence is observable in our updated yearly and monthly climatology (Figures 10, 12, S6, and S7), especially in July. For SED, please see our answers and clarifications in the above.

*In lines 593-594, it would have been good to try to compare your rather large percentage of wind reports being due to derechos and DFs to an estimate of what prior studies showed. I have a feeling your number is much higher, which is consistent with the fact that you are actually studying all bow echoes that produce strong wind, and not true derechos.*

Reply:

Thank you so much for your comments. We made a mistake with the calculation. We averaged the fractions of all sites. Since many sites have a few damaging gusts but the fractions are ~100%, averaging the fractions would overestimate the weight of sites with a few damaging gusts. We now just count the number of damaging gusts regardless of the sites, and the fraction reduces to ~3.0% based on our new dataset. Please see Lines 31 and 648 in the revised main manuscript.

"Additionally, during the study period, derechos account for approximately 3.1% of damaging gust reports ($\geq 25.93$ m s$^{-1}$) reports over the eastern United States."

"On average, DMCSs contribute 4.0% and derechos contribute 3.1% of all damaging gust occurrences."

**References**

Bowman, K. P., & Homeyer, C. R. (2017). *GridRad - Three-Dimensional Gridded NEXRAD WSR-88D Radar Data* the National Center for Atmospheric Research, Computational and Information Systems Laboratory. https://doi.org/https://doi.org/10.5065/D6NK3CR7

CDIACS/EOL/NCAR/UCAR, & CPC/NCEP/NWS/NOAA. (2000). *NCEP/CPC Four Kilometer Precipitation Set, Gauge and Radar* the National Center for Atmospheric Research, Computational and Information Systems Laboratory. https://doi.org/https://doi.org/10.5065/D69Z93M3

Corfidi, S. F., Coniglio, M. C., Cohen, A. E., & Mead, C. M. (2016). A proposed revision to the definition of "derecho". *Bulletin of the American Meteorological Society*, *97*(6), 935-949. https://doi.org/https://doi.org/10.1175/BAMS-D-14-00254.1

Hersbach, H., Bell, B., Berrisford, P., Biavati, G., Horányi, A., Muñoz Sabater, J., Nicolas, J., Peubey, C., Radu, R., Rozum, I., Schepers, D., Simmons, A., Soci, C., Dee, D., & Thépaut, J.-N. (2023). *ERA5 hourly data on single levels from 1940 to present* Copernicus Climate Change Service (C3S) Climate Data Store (CDS). https://doi.org/https://doi.org/10.24381/cds.adbb2d47

Janowiak, J., Joyce, B., & Xie, P. (2017). *NCEP/CPC L3 Half Hourly 4km Global (60S - 60N) Merged IR V1* Goddard Earth Sciences Data and Information Services Center (GES DISC). https://doi.org/https://doi.org/10.5067/P4HZB9N27EKU

---

## Author Comment (AC2)

**Response to Reviewer #2**

Thank you for your careful and thorough reading of the manuscript and your thoughtful comments and suggestions. We apologize for the delay in revising the manuscript, as we spent a significant amount of time on the manual validation and inspection of the identified derechos. According to all three reviewers' comments and suggestions, we have made several significant improvements in our manuscript, which we want to highlight before the point-by-point response.

Firstly, we have changed the wording of some terms so that they are more distinguishable and accurate, and we have also improved the language of many sentences so that the manuscript is more understandable. A derecho is a windstorm, while an MCS is a convective system. They are different concepts. Therefore, we change "derecho" to "derecho-producing MCS" when we refer to the MCS associated with a derecho, and "derecho feature" has been changed to "derecho." The dataset developed in this study includes tracking of both derechos and corresponding derecho-producing MCSs.

Secondly, we have incorporated a "forward propagating" criterion in our derecho detection algorithm. Our initial understanding of the term "forward propagating" was incomplete, and we failed to recognize its critical role in defining a derecho proposed by Corfidi et al. (2016). After careful consideration and evaluation, we have adopted and modified two criteria from Corfidi et al. (2016) to establish the definition of "forward propagating." One is that the acute angle between the averaged bow echo orientation and the bow echo series' propagation direction is larger than 45°, and the other is that the bow echo series' propagation speed is at least 30% faster than the 500-hPa background wind speed. Implementing the "forward propagating" criterion removes many windstorms externally forced by extratropical cyclones, aligning with the purpose of Corfidi et al. (2016), which intends to define derechos as internally driven windstorms. "Externally forced" and "internally driven" reflect distinct physical formation mechanisms of those windstorms, which is why Corfidi et al. (2016) proposed a physically based derecho definition. With the updated detection algorithm, the derecho number between 2004 and 2021 has been substantially reduced from 556 to 274 (for ISD) and 220 (for SED). In addition, due to the inclusion of the "forward propagating" constraint in our derecho algorithm, we have decided not to change the name of our dataset to "high wind-producing bow echo." We have updated all the results in Section 6 based on the improved dataset.

Thirdly, we have developed another parallel dataset using gust speeds from the Storm Events Database (SED). Now, our derecho dataset consists of two subsets: one based on gust measurements from the global hourly Integrated Surface Database (ISD) and the other based on SED gust speeds. Although there are some discrepancies between the two subsets, their agreement is much larger than their difference (Figures 9-12 in the revised main manuscript; or Figure R1 below). Moreover, both ISD and SED gust speeds have limitations and uncertainties, hence differences between the ISD-based and the SED-based datasets are expected and understandable. In addition, it also indicates that our usage of lower gust speed criteria for ISD measurements is reasonable and does not change the derecho number much. We must emphasize that using lower gust speed criteria for ISD measurements than SED reports does not mean that the ISD-identified derechos are weaker than the SED-identified ones (or even not derechos). This

is a compromise, considering that ISD stations are limited and may miss many damaging gusts, as we highlighted in Lines 460-469 in the revised main manuscript (as below).

"We emphasize that, in Criterion 4, our ISD gust speed criteria are weaker than the SED gust speed criteria as well as those of previous studies (Squitieri et al., 2023; Bentley and Mote, 1998; Johns and Hirt, 1987), which also estimated the gust swath based on SED damaging gusts. As mentioned in Section 2.2.2, most SED gust reports are estimates, while ISD provides gust measurements from weather stations. SED estimates can capture potential damaging gust occurrences over a much larger area, although with large uncertainties. In contrast, due to the limited coverage of observational sites, real-time ISD measurements may miss substantial damaging gust occurrences in nearby regions. Therefore, we lower the gust speed criteria to capture potential derechos when using ISD measurements. It does not mean that the ISD-based derechos are weaker than the SED-based ones or even not derechos, as elaborated in Section 5."

Fourthly, due to the incorporation of "forward propagating" in the detection algorithm and the development of the SED-based derecho dataset, we have updated some sensitivity tests, evaluations, and the comparisons of our datasets with the NOAA SPC data in 2004 and 2005 and previous studies. Please see Section 5 for further details.

[Figure]

Figure R1. Bar chart of the annual derecho numbers from the ISD-based and the SED-based datasets from 2004 to 2021. Gray shading denotes derechos captured by both datasets, red shading refers to derechos only identified when using ISD gust observations, and blue shading represents SED-only derechos. The figure is the same as Figure 9 in the revised main manuscript.

*This article describes the development of a machine learning approach to create a derecho climatology across the United States. The novelty and originality of the work should be praised. The authors for the the most part have a well reasoned approach and methodology to creating this dataset, however there are a few major items of concern that stood out during this review:*

*I struggle with understanding which definition of a derecho the authors are using and also relying on to classify a feature as a derecho. In the background/introduction the authors present a history on the evolution of the definition of a derecho. I encourage the authors to keep this in the introduction, but I also encourage to authors to present the definition of a derecho they chose for their methodology clearly and provide additional reasoning on why the this specific definition was chose. The authors should really try to use a definition that most closely represents the official definition used by the National Weather Service and/or Storm Prediction*

*Center. Using a definition that either has a shorter length requirement (or longer one) would impact the number of derechos that are classified in your results.*

Reply:

Thank you for your comments and suggestions. As explained above, we change the wording of "derecho features" and hope the manuscript is more understandable now. In the introduction, we highlight that this study aims to develop a derecho dataset following the definition proposed by Corfidi et al. (2016) (Lines 84-90 and 103-107 in the revised main manuscript, or as below). We did not intend to change anything in Corfidi et al.'s definition, but because of the limitations of the wind gust datasets and uncertainties of bow echo identification, we modified some thresholds and introduced the details of how we apply Corfidi et al.'s definition to our available datasets in Section 4. This is a compromise but not a change of definition. Follow your suggestion, we have rewritten Section 4: we first provide a relatively simple derecho definition (as below) and then explain the details separately.

"Considering the inconsistent thresholds used in the above studies and the lack of physical mechanisms in their derecho definitions, Corfidi et al. (2016) proposed a stricter and more physically based derecho definition, which required the existence of sustained bow echoes with mesoscale vortices or rear-inflow jets and a nearly continuous wind damage swath of at least 100 km wide along most of its extent and 650 km long. In addition, the wind damage must occur after the convective system was organized into a cold-pool-driven forward-propagating MCS."

"This study applies a semantic segmentation convolutional neural network (CNN) to detect bow echoes automatically from two-dimensional composite (column-maximum) reflectivity ($Z_{Hmax}$) data in the United States, which are then combined with an MCS tracking dataset and surface gust speeds to identify derechos using criteria adjusted from Corfidi et al. (2016)."

"Our final criteria are summarized below, with detailed explanations provided afterward.

1) A derecho must be attached to an MCS from the MCS dataset.

2) The derecho must persist for at least 5 hours, with a bow echo present for at least 80% of its lifetime. In addition, gaps between successive bow echo occurrences cannot exceed two hours. All bow echoes must belong to the same bow echo series, as defined in the following section.

3) The derecho bow echo series must exhibit forward propagation, based on two modified criteria from Corfidi et al. (2016):

   • The acute angle between the averaged bow echo orientation and the bow echo series' propagation direction must exceed 45° (Figure 6).

- The propagation speed of the bow echo series must be at least 30% greater than the background mean wind speed at 500 hPa, derived from ERA5 data. The methodology for calculating the averaged bow echo orientation, bow echo series' propagation direction and speed, and the background mean wind speed is detailed in Appendix A.

4) Derecho-associated gust speed criteria vary based on the gust speed source dataset:

- For ISD data: Within 100 km of the derecho-accompanied bow echoes (termed the "derecho area"), there must be at least 10 sites with strong gusts ($\geq 17.43$ m s$^{-1}$) and at least 1 site with damaging gusts ($\geq 25.93$ m s$^{-1}$).

- For ISD data: At least 10 locations must report damaging gusts.

- The fraction of sites with strong/damaging gusts (ISD) or damaging gusts (SED) must be $\geq 20\%$.

- Gaps between successive strong (ISD) or damaging (SED) gust reports cannot exceed two hours.

- The gust swath must be at least 650 km in length and 100 km in width. Swath length and width calculations are explained below."

*I do not understand the inclusion of surface wind speed observations in this manuscript. Derechos are classified operationally through the Storm Events Database (i.e. local storm reports), not through surface wind observations.*

Reply:

Please see above our explanation and clarification.

*The organization of introduction needs quite a bit of improvement as well. It was very difficult to follow in terms of readability, partly compounded by the presentation of all the definitions of derechos. The introduction also presents Figure 1 which is a very very busy figure and in its current form, takes away from the paper. I recommend the authors overhaul the section to provide clarity on previous research, the definition of a derecho and motivation for their great ideas as far as developing this database.*

Reply:

Thank you for your comments and suggestions. We have simplified Figure 1 and reorganized the sentences for the first derecho definition in Lines 59-71 (as below). Also, as mentioned at the beginning of this document highlighting the key changes in the revision, we have changed the wording of some terms and hope these changes can improve the reading of the manuscript.

'Specifically, they required a derecho to satisfy the following six criteria.

1) There must be a concentrated area of reports with wind damage or convective gusts > 25.7 m s$^{-1}$, and the major axis length of the area must be at least 400 km.

2) Those wind damage or convective gust reports must show a pattern of chronological progression, either as a singular swath or a series of swaths.

3) The concentrated area must have at least three reports of either F1 damage (32.7-50.3 m s$^{-1}$) (Fujita, 1971) or convective gust of at least 33.4 m s$^{-1}$ separated by ≥ 64 km.

4) At most 3 hours can elapse between successive reports of wind damage or gust > 25.7 m s$^{-1}$.

5) The associated convective system must have temporal and spatial continuity in surface pressure and wind fields.

6) If multiple swaths of wind damage or gust reports > 25.7 m s$^{-1}$ exist, they must be from the same MCS event.'

*It is difficult to evaluate the results that are presented, especially with the current derecho definition that is used. The current definition that is used (and with sfc wind obs) makes the number of derechos classified by this current form of research difficult to believe. Hopefully an overhaul in the definition used will provide a more realistic number of derechos identified. I do like Figures 9, 10, and 11 in presenting the results. These are great and easy to interpret graphics. I encourage these graphics to stay but adjusted with potential adjustments from the reviews. I would like to see potentially see how the next iteration of these graphics compare to actual confirmed derechos from the same time period.*

Reply:

Thank you for your comments. We want to clarify that we follow the derecho definition proposed by Corfidi et al. (2016), which is different from conventional definitions. Since a derecho climatology based on Corfidi et al.'s definition is unavailable, we can only compare our results with prior studies based on conventional definitions. For the details on how we improve the manuscript, please see our response at the very beginning.

**References**

Corfidi, S. F., Coniglio, M. C., Cohen, A. E., & Mead, C. M. (2016). A proposed revision to the definition of "derecho". Bulletin of the American Meteorological Society, 97(6), 935-949. https://doi.org/https://doi.org/10.1175/BAMS-D-14-00254.1

Fujita, T. T. (1971). Proposed characterization of tornadoes and hurricanes by area and intensity.

---

## Author Comment (AC3)

**Response to Reviewer #3**

Thank you for your careful and thorough reading of the manuscript and your thoughtful comments and suggestions. We apologize for the delay in revising the manuscript, as we spent a significant amount of time on the manual validation and inspection of the identified derechos. According to all three reviewers' comments and suggestions, we have made several significant improvements in our manuscript, which we want to highlight before the point-by-point response.

Firstly, we have changed the wording of some terms so that they are more distinguishable and accurate, and we have also improved the language of many sentences so that the manuscript is more understandable. A derecho is a windstorm, while an MCS is a convective system. They are different concepts. Therefore, we change "derecho" to "derecho-producing MCS" when we refer to the MCS associated with a derecho, and "derecho feature" has been changed to "derecho." The dataset developed in this study includes tracking of both derechos and corresponding derecho-producing MCSs.

Secondly, we have incorporated a "forward propagating" criterion in our derecho detection algorithm. Our initial understanding of the term "forward propagating" was incomplete, and we failed to recognize its critical role in defining a derecho proposed by Corfidi et al. (2016). After careful consideration and evaluation, we have adopted and modified two criteria from Corfidi et al. (2016) to establish the definition of "forward propagating." One is that the acute angle between the averaged bow echo orientation and the bow echo series' propagation direction is larger than 45°, and the other is that the bow echo series' propagation speed is at least 30% faster than the 500-hPa background wind speed. Implementing the "forward propagating" criterion removes many windstorms externally forced by extratropical cyclones, aligning with the purpose of Corfidi et al. (2016), which intends to define derechos as internally driven windstorms. "Externally forced" and "internally driven" reflect distinct physical formation mechanisms of those windstorms, which is why Corfidi et al. (2016) proposed a physically based derecho definition. With the updated detection algorithm, the derecho number between 2004 and 2021 has been substantially reduced from 556 to 274 (for ISD) and 220 (for SED). In addition, due to the inclusion of the "forward propagating" constraint in our derecho algorithm, we have decided not to change the name of our dataset to "high wind-producing bow echo." We have updated all the results in Section 6 based on the improved dataset.

Thirdly, we have developed another parallel dataset using gust speeds from the Storm Events Database (SED). Now, our derecho dataset consists of two subsets: one based on gust measurements from the global hourly Integrated Surface Database (ISD) and the other based on SED gust speeds. Although there are some discrepancies between the two subsets, their agreement is much larger than their difference (Figures 9-12 in the revised main manuscript; or Figure R1 below). Moreover, both ISD and SED gust speeds have limitations and uncertainties, hence differences between the ISD-based and the SED-based datasets are expected and understandable. In addition, it also indicates that our usage of lower gust speed criteria for ISD measurements is reasonable and does not change the derecho number much. We must emphasize that using lower gust speed criteria for ISD measurements than SED reports does not mean that the ISD-identified derechos are weaker than the SED-identified ones (or even not derechos). This

is a compromise, considering that ISD stations are limited and may miss many damaging gusts, as we highlighted in Lines 460-469 in the revised main manuscript (as below).

"We emphasize that, in Criterion 4, our ISD gust speed criteria are weaker than the SED gust speed criteria as well as those of previous studies (Squitieri et al., 2023; Bentley and Mote, 1998; Johns and Hirt, 1987), which also estimated the gust swath based on SED damaging gusts. As mentioned in Section 2.2.2, most SED gust reports are estimates, while ISD provides gust measurements from weather stations. SED estimates can capture potential damaging gust occurrences over a much larger area, although with large uncertainties. In contrast, due to the limited coverage of observational sites, real-time ISD measurements may miss substantial damaging gust occurrences in nearby regions. Therefore, we lower the gust speed criteria to capture potential derechos when using ISD measurements. It does not mean that the ISD-based derechos are weaker than the SED-based ones or even not derechos, as elaborated in Section 5."

Fourthly, due to the incorporation of "forward propagating" in the detection algorithm and the development of the SED-based derecho dataset, we have updated some sensitivity tests, evaluations, and the comparisons of our datasets with the NOAA SPC data in 2004 and 2005 and previous studies. Please see Section 5 for further details.

[Figure]

Figure R1. Bar chart of the annual derecho numbers from the ISD-based and the SED-based datasets from 2004 to 2021. Gray shading denotes derechos captured by both datasets, red shading refers to derechos only identified when using ISD gust observations, and blue shading represents SED-only derechos. The figure is the same as Figure 9 in the revised main manuscript.

*Major Comments*
*This paper presents a novel, machine-learning scheme to objectively identify derecho-producing convective systems. The paper reads well and, not withstanding the minor comments indicated below, to this reviewer seems both well-organized and well-presented. I have only two main comments.*
*First, being familiar with the vagaries of the severe-weather report database, I support the authors' use of the Integrated Surface Database (ISD). While the ISD arguably is subject to its own limitations, the quality-control algorithms used offer a higher and more universal level of uniformity than that associated with the severe-weather report database. The large number of derecho-producing convective systems identified with the current approach of the authors compared with those of previous studies largely reflects the rather low wind threshold employed; use of a somewhat higher threshold (and/or duration threshold) would lower the number of*

*events identified. Obviously the true frequency of derecho-producing events remains unknown; the lower frequencies suggested by previous studies may be somewhat low.*

Reply:

We sincerely appreciate your support for our usage of the ISD. To make the study more reliable and robust, and in response to the other reviewers, we have developed an additional dataset based on SED. Comparison between the two datasets justifies the usage of lower wind thresholds for the ISD. The overestimation in our initial manuscript is mainly due to incomplete implementation of the "forward propagating" criterion from Corfidi et al. (2016). Revising the "forward propagating" in our detection algorithm removes many externally forced systems. For more details, please see the highlights of our responses above.

*The "true frequency" point brings into mind the main purpose of the present study --- objective identification of derecho-producing MCSs. The number of systems identified is sensitive to the underlying definition used in the scheme. This is where difficulty has arisen in the past and to some extent continues with the present paper. The omission of "forward propagating" from the current definition of the authors (page 516 ff) is problematic. Sustained forward-propagation is a fundamental aspect of derecho-producing convective systems. In absence of such a criterion one could argue that the approach described in the present paper is closer to that of a bow-echo detection scheme. Derecho-producing convective systems could be described as arising from bow-echo producing processes --- including rapid, sustained forward propagation --- that remain active for extended periods of time. I suggest re-visiting the abortive attempts made (lines 517-519) to identify the presence of forward propagation and refine the ML approach used here.*

Reply:

Thank you so much for the suggestions. As mentioned above, our initial understanding of "forward propagating" is incomplete. We thought it was just a direction metric. After careful reading of Corfidi et al. (2016), we realize the importance of this criterion and its relationship with physical processes. After revising the "forward propagating" criterion (Lines 381-388 in the revised main manuscript), the dataset quality has been much improved.

"The derecho bow echo series must exhibit forward propagation, based on two modified criteria from Corfidi et al. (2016):

- The acute angle between the averaged bow echo orientation and the bow echo series' propagation direction must exceed 45° (Figure 6).

- The propagation speed of the bow echo series must be at least 30% greater than the background mean wind speed at 500 hPa, derived from ERA5 data. The methodology

for calculating the averaged bow echo orientation, bow echo series' propagation

direction and speed, and the background mean wind speed is detailed in Appendix A."

*Minor comments (numbers refer to line numbers in version of 24 June 2024)*
*40. Change "magnitude" to "impact"*
Reply:

Corrected. Thank you!

*190. Consider adding a parenthetical description of "skip connections"*
Reply:

Added. Thank you!

"Dense Nets are notable for their large number of skip connections (which create multiple paths for data to flow through the network without passing through every layer), and they can achieve comparable performance to very large classifier CNNs with only a fraction of the trainable parameters."

*201. Clarify what is meant by "more distinct"*
Reply:

The initial 556 positive samples are from 54 named derechos. The new negated 500 positive samples may be from 100 convective systems. Here, "more distinct" just refers to the samples from diverse "sources" to improve its representation.

*267. Define or reference "binary cross entropy loss"*
Reply:

A reference added. Thanks.

"It is trained using binary cross entropy loss (Bishop, 2006) on masks generated from its 384, 192, 96, 48, 24, and 12-pixel resolution feature representations (Huang et al., 2020), though only the full-resolution (384×384-pixel) output mask is used at inference time."

*307-310. Well-stated*
Reply:

Thank you!

*312. Not completely sure what is meant by the "upper" and "lower" parts of the table*
Reply:

Corrected to "upper triangular" and "lower triangular". Thank you!

*366. Add parenthetically, "Derecho feature" after "DF"*

Reply*:*

Thank you for your suggestion. We change the wording of "derecho feature" to "derecho" throughout the manuscript.

*371-372. Not sure that this criterion would always be helpful...*
Reply*:*

It is not always helpful. But it does help remove some extratropical cyclones. We manually check every identified derecho and high-wind-producing convective system to ensure the criterion does not produce any adverse impact.

*385. Makes sense!*
Reply*:*

Thank you! By comparing our SED-based and ISD-based subsets, we find lowering the gust speed criterion is reasonable for the ISD.

*433-434. Agree with this focus; consider also maintaining the forward-propagating aspect.*
Reply*:*

Thank you for your suggestion. The sentence has been deleted since we rewrite the evaluation section (Section 5) due to the incorporation of forward-propagating and the inclusion of SED.

*455. Should "as" be "than"?*
Reply*:*

The sentence has been deleted. Thank you.

*480. Consider parenthetically adding "cold-cloud shield" given that acronym has not been used since line 129.*
Reply*:*

Corrected. Thank you!

"The 50% areal overlap threshold in PyFLEXTRKR, which links consecutive cold cloud shields (CCSs), may fail to capture very fast-moving convective systems using hourly satellite and NEXRAD data."

*520. The addition of a simple schematic to illustrate the angles mentioned would be helpful*
Reply*:*

Thank you for the suggestion. We have updated the algorithm schematic (as Figure R2 below) to illustrate the angle. Besides, in Appendix A, we provide more details on how we calculate them.

[Figure]

Figure R2. Schematic of the automated detection algorithm.

*525-526. Good to see this explicitly stated*
Reply:

Thank you for positive comment. Unfortunately, our understanding of "forward propagating" was incomplete. We remove the sentence and incorporate "forward propagating" in the derecho detection algorithm.

*556. Should "west-east" be "northwest-southeast"?*
Reply:

The sentence is deleted. We remove this sensitivity test, which is unnecessary after the development of the SED-based dataset.

*588. Should "2014" in Figure 12 caption be "2004"?*
Reply:

Corrected. Thank you for pointing out the error.

*676. Capitalize "Weather" and "Review"*
Reply:

Corrected. Thank you!

*712. Add publication in which this manuscript appeared*
Reply*:*

This is a report, and we add the website of the report.

*731. Add year of publication (2020 (?), per line 234)*
Reply*:*

Corrected. Thank you!

*738. Capitalize "Python"*
Reply*:*

Corrected. Thank you!

*791. Capitalize "Atmospheric Sciences"*
Reply*:*

Corrected. Thank you!

**References**

Corfidi, S. F., Coniglio, M. C., Cohen, A. E., & Mead, C. M. (2016). A proposed revision to the definition of "derecho". Bulletin of the American Meteorological Society, 97(6), 935-949. https://doi.org/https://doi.org/10.1175/BAMS-D-14-00254.1

Huang, H., Lin, L., Tong, R., Hu, H., Zhang, Q., Iwamoto, Y., Han, X., Chen, Y.-W., & Wu, J. (2020). Unet 3+: A full-scale connected unet for medical image segmentation. ICASSP 2020-2020 IEEE international conference on acoustics, speech and signal processing (ICASSP),

---

## Referee Report (RR1)

This paper has been much improved by the revisions. I applaud the authors for including an analysis using SED reports, and for very helpful figures like Fig. 6. It is also nice to see the change in criteria to match better with current definitions of derechos.

Despite the many improvements, I do still have several major concerns that require revisions to the paper. The first one below is particularly serious, but understandable since this happens often when one paper is undergoing revision while another that is very similar by different authors suddenly is published. In the present situation, that paper is so closely related to your own that you will want to not only refer to it, but also change a comparison that you currently make to a rather different dataset (more of an apples to oranges comparison) to instead compare to the new data in the new paper (much more of an apples to apples comparison). The major revisions I request are listed below:

1. One big problem is that although you did include some references to the Part 1 paper by Squitieri and coauthors in 2023 based on my earlier review, you have failed to recognize that Part 2 has already been published (2025, BAMS). That paper has some of the same type of analyses that you do (spatial analyses) using the new definition of a derecho hinted at in the 2023 paper and following Corfidi (2016) as you have done. Clearly, it is very important that you add comparison to this other study now that it has been published. Your paper makes an excellent companion piece to that one, since there is now a lot of similarity, which allows you to truly do a precise comparison and establish how well AI may be working. My own quick glance shows they find very few derechos per year, ranging from 1 to 8 per year. It seems to me that you should replace your comparison with an old 2 years of SPC data (50 cases) with a comparison to the new Squitieri et al. (2024) paper, which is a far better apples to apples comparison with your definition of a derecho. In addition, the really small number of derechos they now find in the US per year raises some questions about some of your tunable parameters. In my third major comment below, I talk about how you should consider your overestimate of ISD-only derechos as evidence you have dropped your threshold too low. In light of the Squitieri et al. (2025) paper, an increase in that threshold would not only result in better agreement between your SED- and ISD-determined number of derechos, but would also likely lower your totals, matching a bit better the small numbers found in the 2025 paper.

2. At line 414 when you are describing your criteria, how do you determine a fraction of SED sites above 20%? I can understand with the other dataset of surface observations that you must look at the total number of stations falling within your derecho swath, and you can figure out what 20% of those stations are. However, for the SED reports, there is no grand total to use. Reports come in from whoever bothers to make the report. Near a city, many reports can come in close proximity to each other, while in wide open areas with low population, reports can be few and far between. There is no total number of stations, so it is impossible to compute a percentage or fraction. More explanation is needed here because I have no idea how you can even attempt to apply this criterion to SED reports.

3. In the discussion from lines 633-645, to me, that agreement does not seem very good. I agree it is not terrible, but the union of all 3 possibilities is 322 cases. 172, or about 55%, are matched

with the use of both datasets.  You admit that you used an arbitrary adjustment downward to set a threshold for your ISD cases.  Wouldn't this discussion here have been a really good starting point to explore what happens with different ISD thresholds?  I would think a good scientific approach would have been to use the threshold in an iterative manner to get the best match.  If you are truly correcting for the lack of ISD stations by using a lower threshold, wouldn't it be best to have found a threshold that results in the largest intersection with the most cases showing up in both datasets and the least number of cases only showing up in one or the other?  This seems like relatively low-hanging fruit.  Instead, you did not do any sensitivities to the threshold (I do realize you said your general behavior between the cases is relatively similar) but since you seem to be arguing that this is a product the community can already go use, I would have thought it would be very important to at least try to get the best match possible.  For some of the reasons you mentioned regarding errors in SED, you will probably never be able to eliminate those cases that show up there but do not show up in ISD.  However, this was a substantially smaller number than what you found showing up in ISD but not SED.  That overestimate in ISD cases is surely linked to having the bar set too low for your wind threshold.  What would happen if you raised your arbitrary threshold by 2.5 m/s?  Your discussion of sensitivities later on in the paper is useful, but you do not mention this particular sensitivity, which seems the most correctable of all of them to me. You know that you are arbitrarily pulling a number out of the air, smaller than the usually used severe threshold, because you correctly point out that a smaller threshold should be justified due to a lack of stations.  But there is nothing we can point to in science to know exactly how much below the original threshold you should be.  Wouldn't a fine tuning of the numbers right in this discussion have been a good way to determine the best threshold to have adjusted downward to?

The following are other minor revisions that are needed:

1. Change line 114 from "this study" to "The present study".  We cannot tell from "this" if you mean your study or one of the ones you just mentioned.

2. Lines 233-234, what do you mean by a negative sample?  Is this a bow echo that does not have damaging wind associated with it during the hour? Please provide more details. In this paragraph, it sounds like you are only looking at 54 known derechos, and hourly images.  It seems like 566 positive and 4000 negative are far too many if you are just using hourly images for 54 events.

3. In Figure 12 and the discussion around it, I am confused. What is the difference between wind reports in an DMCS versus a derecho?  I thought a DMCS is an MCS that produces a derecho.  Is the information in panel c simply wind reports that happen in an MCS that did have a derecho, but falling outside the location or time window that you define to be a derecho?  I don't feel like this is very clear in the paper, and believe when you start showing statistics relating to these different types of systems, you need to remind the reader what the difference is, or in this case, how you can have differences in reports from DMCSs compared to derechos.

---

## Author Response (AR2)

Dr. Graciela Raga
Editor
ESSD
April 8, 2025

Subject: Revision of manuscript # essd-2024-112

Dear Dr. Raga,

Thank you and the reviewers for your thoughtful and thorough review of our manuscript. In addition to comments from Reviewer #1, we also received feedback from Reviewer #3 via email. Reviewer #3 was unable to meet the submission deadline and upload comments to the ESSD system. However, we have nonetheless carefully considered their input alongside the other reviewer's remarks. Additionally, we have made further minor corrections to the manuscript to ensure it meets the publication standards.

Please direct any correspondence regarding this manuscript to me (jianfeng.li@pnnl.gov) or Dr. Zhe Feng (zhe.feng@pnnl.gov). We sincerely appreciate your time and effort throughout this process.

Sincerely,
Jianfeng Li
Atmospheric, Climate, and Earth Sciences Division
Pacific Northwest National Laboratory
Richland, Washington, US, 99354

**Response to Reviewer #1**

Thank you for your careful and thorough reading of this manuscript and your thoughtful comments and suggestions. Our responses follow the reviewer's comments (*in Italics*).

*This paper has been much improved by the revisions. I applaud the authors for including an analysis using SED reports, and for very helpful figures like Fig. 6. It is also nice to see the change in criteria to match better with current definitions of derechos.*

*Despite the many improvements, I do still have several major concerns that require revisions to the paper. The first one below is particularly serious, but understandable since this happens often when one paper is undergoing revision while another that is very similar by different authors suddenly is published. In the present situation, that paper is so closely related to your own that you will want to not only refer to it, but also change a comparison that you currently make to a rather different dataset (more of an apples to oranges comparison) to instead compare to the new data in the new paper (much more of an apples to apples comparison). The major revisions I request are listed below:*

**Reply:**

Thank you for reviewing the manuscript carefully and thoroughly again. We sincerely appreciate your efforts to help improve the manuscript. We have addressed all your comments below.

*1. One big problem is that although you did include some references to the Part 1 paper by Squitieri and coauthors in 2023 based on my earlier review, you have failed to recognize that Part 2 has already been published (2025, BAMS). That paper has some of the same type of analyses that you do (spatial analyses) using the new definition of a derecho hinted at in the 2023 paper and following Corfidi (2016) as you have done. Clearly, it is very important that you add comparison to this other study now that it has been published. Your paper makes an excellent companion piece to that one, since there is now a lot of similarity, which allows you to truly do a precise comparison and establish how well AI may be working. My own quick glance shows they find very few derechos per year, ranging from 1 to 8 per year. It seems to me that you should replace your comparison with an old 2 years of SPC data (50 cases) with a comparison to the new Squitieri et al. (2024) paper, which is a far better apples to apples comparison with your definition of a derecho. In addition, the really small number of derechos they now find in the US per year raises some questions about some of your tunable parameters. In my third major comment below, I talk about how you should consider your overestimate of ISD-only derechos as evidence you have dropped your threshold too low. In light of the Squitieri et al. (2025) paper, an increase in that threshold would not only result in better agreement between your SED- and ISD-determined number of derechos, but would also likely lower your totals, matching a bit better the small numbers found in the 2025 paper.*

**Reply:**

Thank you so much for pointing out to us about the publication of the new papers (Squitieri et al., 2025a, b), which are directly related to our study. We have added a comparison between our study and Squitieri et al. (2025b) (Lines 519-526 in the revision-tracked main manuscript) and cited the paper a few more times when describing the derecho climatological characteristics (Lines 589-592, 618-619, 622, and 633-637). However, we would like to keep the comparison

against the NOAA SPC data, which provides the exclusive event-scale information. The total derecho number comparison cannot guarantee that our dataset captures the correct "derecho" events even if the total counts are similar. The derecho dataset developed in Squitieri et al. (2025b) was not made available publicly, preventing an event-based comparison as what we have done with the NOAA SPC data. We acknowledge the limitation of comparing our dataset with the NOAA SPC data, but it is critical to show that our detection algorithm identifies the "correct" events. As noted in the response below, differences between Squitieri et al. (2025b) and our dataset may be attributed to many reasons, suggesting a detailed event-based comparison combined with sensitivity analysis would be useful in the future. For questions related to ISD, please refer to our responses to the third major comment.

Lines 519-526: 'Our derecho numbers are also higher than those from Squitieri et al. (2025b), who identified 70 SED-based derechos during 2000-2022 based on the physically-based definition criteria from Corfidi et al. (2016) but with much stricter gust requirements (e.g., at least five reports of very damaging gusts ($\geq 33.53$ m s$^{-1}$)) for a 400-km-long gust swath (Squitieri et al., 2025a, b). The discrepancies among the present study, Corfidi et al. (2016), and Squitieri et al. (2025b) could be attributed to the different gust criteria used in the derecho definitions but also likely stem from differences in the methods used to calculate gust swath length and width, the criteria for forward propagation, and the diverse observational source datasets used in the derecho detection.'

Lines 589-592: 'While if we require at least five very damaging gust reports when using SED, the derecho count decreases substantially from 220 to 125, which is still larger than but much closer to the estimates by Squitieri et al. (2025b) (70 derechos between 2000 and 2022).'

Lines 634-636: 'However, our dataset has almost no derechos in the cold seasons, which is generally not the case in previous studies except for Squitieri et al. (2025b), which also used physically-based criteria to detect derechos.'

*2. At line 414 when you are describing your criteria, how do you determine a fraction of SED sites above 20%? I can understand with the other dataset of surface observations that you must look at the total number of stations falling within your derecho swath, and you can figure out what 20% of those stations are. However, for the SED reports, there is no grand total to use. Reports come in from whoever bothers to make the report. Near a city, many reports can come in close proximity to each other, while in wide open areas with low population, reports can be few and far between. There is no total number of stations, so it is impossible to compute a percentage or fraction. More explanation is needed here because I have no idea how you can even attempt to apply this criterion to SED reports.*

**Reply:**

Thank you for your comments. Yes, the 20% criterion is primarily used for ISD gust speeds, and SED seldom reports gust speeds weaker than strong gusts. Implementing the 20% criterion for SED data excludes only one MCS from being considered a potential DMCS, which we can also remove by manually correcting the falsely identified bow echoes, although the latter is more time-consuming. However, to keep the criteria consistent between ISD and SED as much as possible, we still apply the 20% criterion to SED data, which does not produce any adverse impact on the derecho detection according to our manual check of all identified derechos and high-wind-producing convective systems.

The implementation of the 20% criterion for SED data is rather straightforward. Each SED gust report has a location (latitude and longitude), as mentioned in Lines 194-198, and we consider the location a "site". Then we can count the number of unique "sites" with damaging gusts or gust speeds weaker than damaging gusts.

We have added a sentence in Lines 445-447 to make the 20% criterion more understandable.

"It is noteworthy that this criterion is primary applicable to ISD data, and its implementation for SED data excludes only one MCS from being considered a potential DMCS."

*3. In the discussion from lines 633-645, to me, that agreement does not seem very good. I agree it is not terrible, but the union of all 3 possibilities is 322 cases. 172, or about 55%, are matched with the use of both datasets. You admit that you used an arbitrary adjustment downward to set a threshold for your ISD cases. Wouldn't this discussion here have been a really good starting point to explore what happens with different ISD thresholds? I would think a good scientific approach would have been to use the threshold in an iterative manner to get the best match. If you are truly correcting for the lack of ISD stations by using a lower threshold, wouldn't it be best to have found a threshold that results in the largest intersection with the most cases showing up in both datasets and the least number of cases only showing up in one or the other? This seems like relatively low-hanging fruit. Instead, you did not do any sensitivities to the threshold (I do realize you said your general behavior between the cases is relatively similar) but since you seem to be arguing that this is a product the community can already go use, I would have thought it would be very important to at least try to get the best match possible. For some of the reasons you mentioned regarding errors in SED, you will probably never be able to eliminate those cases that show up there but do not show up in ISD. However, this was a substantially smaller number than what you found showing up in ISD but not SED. That overestimate in ISD cases is surely linked to having the bar set too low for your wind threshold. What would happen if you raised your arbitrary threshold by 2.5 m/s? Your discussion of sensitivities later on in the paper is useful, but you do not mention this particular sensitivity, which seems the most correctable of all of them to me. You know that you are arbitrarily pulling a number out of the air, smaller than the usually used severe threshold, because you correctly point out that a smaller threshold should be justified due to a lack of stations. But there is nothing we can point to in science to know exactly how much below the original threshold you should be. Wouldn't a fine tuning of the numbers right in this discussion have been a good way to determine the best threshold to have adjusted downward to?*

**Reply:**

Thank you very much for your comments and suggestions. Firstly, we want to emphasize that both ISD and SED datasets have limitations, and the derecho datasets developed based on the two gust speed datasets inherit those limitations. Neither of the derecho datasets nor their combination can be considered the ground truth. An overlap of 172 derechos between the ISD-based (274 derechos) and SED-based (220 derechos) datasets is imperfect but may be acceptable considering various uncertainties involved in developing the derecho datasets. Secondly, we acknowledge that we do not have a robust scientific reason to select 17.43 m s$^{-1}$ as the ISD gust speed threshold to detect derechos. There are a few named gusts from the National Weather Service, including violet gust ($> 41.13$ m s$^{-1}$), very damaging gust ($\geq 33.53$ m s$^{-1}$), damaging gust ($\geq 25.93$ m s$^{-1}$), and strong gust ($\geq 17.43$ m s$^{-1}$). We selected the strong gust speed as the

threshold for ISD because it ranks right below damaging gust. The threshold is undoubtedly adjustable. Following your suggestions, we have performed several sensitivity tests with different ISD gust speed thresholds. The derecho number is sequentially reduced from 274 when using a threshold of 17.43 m s$^{-1}$ to 255, 229, 210, and 157 for a threshold of 18, 18.5, 19, and 20 m s$^{-1}$, respectively. Even if the total derecho numbers are very close between the ISD-based and SED-based datasets when we use a threshold of 18.5 m s$^{-1}$ for ISD, their overlap reduces to 152 (Figure R1), suggesting the challenge of simultaneously matching the total derecho numbers and increasing the number of overlapping derechos derived from the two datasets. This is within our expectation. The difference between the ISD-based and SED-based derecho datasets stems from the inherent limitations of the ISD and SED gust speed datasets, which cannot be eliminated by simply tuning the gust speed threshold. Per your request, we have added the sensitivity test results in Lines 586-589 in the revised main manuscript, as follows.

'For example, to reduce the ISD-based derecho count to the SED-based level, we must increase the ISD gust speed threshold in Criterion 4 in Section 4.1 from 17.43 m s$^{-1}$ to 18.5 m s$^{-1}$; using the latter threshold produces a derecho number of 229, 152 of which overlapped with the SED-based derecho dataset.'

However, we do not plan update our ISD derecho dataset using the new threshold (18.5 m s$^{-1}$) due to two reasons: 1) 18.5 m s$^{-1}$ does not correspond to any named gusts; 2) the lack of a ground truth to support any fine tuning. For users who are interested in derechos that produce stronger gusts, we encourage them to adjust those thresholds based on their specific scientific needs. As we highlighted in Lines 596-599, to make our dataset useful to the broad users, we store all the key parameters in the dataset to make it flexible; it is straightforward to further tune those thresholds per the users' specific needs.

[Figure]

Figure R1. Same as Figure 8 but using a gust speed threshold of 18.5 instead of 17.43 m s$^{-1}$ for ISD.

*The following are other minor revisions that are needed:*

*1. Change line 114 from "this study" to "The present study". We cannot tell from "this" if you mean your study or one of the ones you just mentioned.*

**Reply:**

Changed. Please see Line 103 in the revised manuscript. Thank you!

"The present study applies a semantic segmentation convolutional neural network (CNN) to detect bow echoes automatically from two-dimensional composite (column-maximum) reflectivity ($Z_{Hmax}$) data in the United States"

*2. Lines 233-234, what do you mean by a negative sample? Is this a bow echo that does not have damaging wind associated with it during the hour? Please provide more details. In this paragraph, it sounds like you are only looking at 54 known derechos, and hourly images. It seems like 566 positive and 4000 negative are far too many if you are just using hourly images for 54 events.*

**Reply:**

Since our machine learning approach is used to identify bow echoes, positive and negative samples only differ in the existence of bow echoes. However, since the negative samples are randomly selected from the entire radar reflectivity dataset, few negative samples may contain bow echoes. We must emphasize that the few "bad" negative samples have minimal impact on the training of a Dense Net (Section 3.1.2), and moreover, this is just the first step of our machine learning part, and we only used Dense Net to collect more new positive samples due to its high false positive rate, as mentioned in Lines 233-237.

It is noteworthy that the initial 566 positive samples are from the 54 named DMCSs, while the negative samples are randomly selected from the entire radar reflectivity dataset embedded in the MCS dataset (18 years of hourly data). As we mentioned in the first round of revision, the mean derecho lifetime is ~11 hours, it is not hard to find 566 positive samples from 54 named DMCSs. The initial 566 positive samples had been validated independently by two of the coauthors and have high qualities. However, we understand the limitation of the initial positive samples, which are from a limited number of derechos and may not be representative. Therefore, we generated a much more diverse positive and negative samples through Dense Net and pseudo-labeling, and the final samples include 1699 positive and 1978 negative samples, as explained in Sections 3.1.2 and 3.1.3.

We have added more information in Lines 219-222 and hope it is more understandable.

"The number of bow echo samples varies among different DMCSs, and 566 positive samples (with bow echoes) are obtained in total. 5400 negative samples (generally without bow echoes) are also randomly selected from the entire 18 years of radar reflectivity data embedded in the MCS dataset."

*3. In Figure 12 and the discussion around it, I am confused. What is the difference between wind reports in an DMCS versus a derecho? I thought a DMCS is an MCS that produces a derecho. Is the information in panel c simply wind reports that happen in an MCS that did have a derecho, but falling outside the location or time window that you define to be a derecho? I don't feel like this is very clear in the paper, and believe when you start showing statistics relating to these different types of systems, you need to remind the reader what the difference is, or in this case, how you can have differences in reports from DMCSs compared to derechos.*

**Reply:**

Thank you for your comments. We have clarified the discussion around Figure 12 in Lines 659-661, as follows. The DMCS damaging gust reports contains both the derecho associated damaging gust reports and those falling outside. Therefore, Figure 12d is a subset of Figure 12c.

"Notably, damaging gust reports associated with a DMCS include those from the corresponding derecho as well as those falling outside the derecho location or time window."

**Reference:**

Squitieri, B. J., Wade, A. R. and Jirak, I. L.: On a modified definition of a derecho. Part I: Construction of the definition and quantitative criteria for identifying future derechos over the

contiguous United States, Bulletin of the American Meteorological Society, 106(1), E84-E110, https://doi.org/10.1175/BAMS-D-24-0015.1, 2025a.

Squitieri, B. J., Wade, A. R. and Jirak, I. L.: On a modified definition of a derecho. Part II: An updated spatial climatology of derechos across the contiguous United States, Bulletin of the American Meteorological Society, 106(1), E111-E124, https://doi.org/10.1175/BAMS-D-24-0140.1, 2025b.

**Response to Reviewer #2**

Thank you for your careful and thorough reading of this manuscript and your thoughtful comments and suggestions. Our responses follow the reviewer's comments (*in Italics*).

*Review of the most revised (dated 7 February 2025) version of ESSD manuscript 2024-112 (numbers refer to line numbers in the 7 February 2025 manuscript).*

*92. Consider adding Corfidi et al. 2016 as a reference at the end of line.*

**Reply:**

Added. Thank you. Please see Lines 93-94 in the revision-tracked main manuscript.

*221. "Timings" is vague; is the indicated time that of the "bow echo" output?*

**Reply:**

It refers to the occurrence time of the bow echo. We have rewritten the sentence as follows (Line 225).

"The subplot titles indicate the bow echo occurrence times."

*239. Likewise, "distinct" is vague; what is meant by "distinct"? Easily recognizable?*

**Reply:**

Here, "distinct" refers to "diverse" or "a broader range of". We meant that the updated 500 positive samples are from a broader range of "sources" to improve their representation. We have changed the sentence as follows (Lines 242-243).

"these samples have higher diversity than the initial bow echoes generated from the named derechos on Wikipedia because they are drawn from a broader range of events"

*256. I'm not sure how "probability of 0.1" applies here; a brief phrase to explain would be helpful.*

**Reply:**

Thank you for your comment. A probability of 0.1 corresponds to 10% of the pixels. For each pixel, the probability to add random noise to it is 0.1. We have clarified it in Line 260, as follows.

"First, random salt and pepper noise is added to 10% of the pixels in each sample with a probability of 0.1 (i.e., it has a probability of 0.1 to add random noise to a pixel)."

*370-400.  Great to see the modifications that were made here compared with what appeared in the first version of the manuscript --- especially the method used to identify forward propagation (i.e., lines 381-388).*

**Reply:**

Thank you for your positive comment!

*402.  Good update to the figure (cf. the figure in the first version of the manuscript)!*

**Reply:**

Thank you!

*608.  Does figure S6 really show this? "SED" is not mentioned in the caption for Figure S6 in my version of the Supplement.*

**Reply:**

Please find below the updated Figure S6 in the first round of revision. Compared to Figure 9 in the main manuscript, the northwest-southeast axis is more noticeable.

[Figure]

Figure S6. Same as Figure 9 but for the SED-based dataset.

*616. "Code" should be "cool."*

**Reply:**

Thank you very much for pointing out the error. We have changed "code" to "cold". Please see Lines 634-635 in the revision-tracked main manuscript.

*626. The axes also may represent two distinct subsets or populations of progressive (forward-propagating) derechos that move (1) primarily west to east vs. (2) those that move more north to south.*

**Reply:**

Thank you so much for pointing out the issue. Based on our most recent self-organization map analysis, the two axes indeed correspond to different types of derechos associated with different large-scale meteorological patterns. We have corrected the sentences as follows (Lines 646-650).

"The axes may represent two distinct types of progressive derechos associated with different large-scale meteorological patterns."

*630. Ditto first sentence in comment made for line 608.*

**Reply:**

Please find below the updated Figure S7 in the first round of revision.

[Figure]

Figure S7. Same as Figure 11 but for SED-based derechos.

*643. Is the meaning of "PDF" introduced prior to this point? If not, I suggest spelling out "probability distribution function."*

**Reply:**

Corrected on Line 666. Thank you!

*664. My copy of the Supplement references a Figure 14 which does not exist in the revised version of the main manuscript.*

**Reply:**

Yes, the supplement has been updated in the first round of revision.

*Your derecho dataset, based on a method that combines machine learning identification of relevant radar features with a physically-based definition of a derecho, should be a valuable resource for the meteorological community.*

**Reply:**

Thank you. We greatly appreciate your kind acknowledgment.